# Annexin A6 mediates calcium-dependent exosome secretion during plasma membrane repair

**Justin Krish Williams[1†], Jordan Matthew Ngo[1†], Isabelle Madeline Lehman[1], Randy Schekman[2*]**

[1]Department of Molecular and Cell Biology, University of California, Berkeley, Berkeley, United States; [2]Department of Molecular and Cell Biology, Howard Hughes Medical Institute, University of California, Berkeley, Berkeley, United States

**Abstract** Exosomes are an extracellular vesicle (EV) subtype that is secreted upon the fusion of multivesicular bodies (MVBs) with the plasma membrane. Exosomes may participate in intercellular communication and have utility as disease biomarkers; however, little is known regarding the physiological stimuli that induce their secretion. $Ca^{2+}$ influx promotes exosome secretion, raising the possibility that exosomes are secreted during the $Ca^{2+}$-dependent plasma membrane repair of tissues damaged by mechanical stress in vivo. To determine whether exosomes are secreted upon plasma membrane damage, we developed sensitive assays to measure exosome secretion in intact and permeabilized cells. Our results suggest that exosome secretion is coupled to $Ca^{2+}$-dependent plasma membrane repair. We find that annexin A6 (ANXA6), a well-known plasma membrane repair protein, is recruited to MVBs in the presence of $Ca^{2+}$ and required for $Ca^{2+}$-dependent exosome secretion, both in intact and in permeabilized cells. ANXA6 depletion stalls MVBs at the cell periphery, and ANXA6 truncations localize to different membranes, suggesting that ANXA6 may serve to tether MVBs to the plasma membrane. We find that cells secrete exosomes and other EVs upon plasma membrane damage and propose that repair-induced secretion may contribute to the pool of EVs present within biological fluids.

**\*For correspondence:**
schekman@berkeley.edu

[†]These authors contributed equally to this work

## Editor's evaluation

This compelling study brings together two earlier observations: that $Ca^{2+}$ influx can trigger exosome release from multivesicular bodies, and that plasma membrane repair after wounding requires $Ca^{2+}$ and involves $Ca^{2+}$-binding annexin proteins. This important work takes these earlier findings in an interesting new direction by showing that exosome release from MVBs is also triggered by $Ca^{2+}$ influx during plasma membrane wounding and requires the annexin isoform ANX6. The study raises the interesting possibility that cell injury and repair may contribute to the release of exosomes into biological fluids.

## Introduction

Protein secretion is an essential process that prokaryotes and eukaryotes utilize for intercellular communication. In eukaryotic cells, most secretory proteins are exported by the conventional secretory pathway which consists of co-translational signal peptide recognition by the signal recognition particle, polypeptide translocation through the Sec61 translocon, and anterograde transport from the endoplasmic reticulum (ER) to the Golgi apparatus via COPII vesicles (*Park and Rapoport, 2012*; *Shan and Walter, 2005*; *Zanetti et al., 2011*). However, eukaryotic cells have also developed alternative

secretory processes that bypass the conventional pathway (*Malhotra, 2013*). Some well-documented examples include the secretion of leaderless cargoes (proteins that lack a signal peptide), such as interleukin-1β, and the egress of cytoplasmic proteins and RNAs within extracellular vesicles (EVs; *Colombo et al., 2014*; *O'Brien et al., 2020*; *Zhang et al., 2020*).

EVs are membrane-enclosed compartments that are exported to the extracellular milieu of cells in culture and in vivo. Eukaryotic cells secrete EV subpopulations that can be classified broadly into two major categories with distinct biogenesis pathways: microvesicles and exosomes (*Colombo et al., 2014*; *van Niel et al., 2018*). Microvesicles form by outward budding from the plasma membrane (*Cocucci et al., 2009*; *Tricarico et al., 2017*). Exosomes are 30–150 nm vesicles formed by inward budding of the limiting membrane of late endosomes to produce multivesicular bodies (MVBs) that contain intraluminal vesicles (ILVs). Upon fusion of MVBs with the plasma membrane, ILVs are released to the extracellular space as exosomes (*Harding et al., 1983*). Exosomes have potential utility as biomarkers for disease progression because they contain protein and small RNA molecules specific to their cell type of origin (*Driedonks and Nolte-'t Hoen, 2018*; *Shurtleff et al., 2017*; *Upton et al., 2021*).

Little is known about the physiological circumstances that could elevate exosome secretion. Interestingly, elevation of cytosolic $Ca^{2+}$ with ionophores has been demonstrated to induce rapid exosome secretion in a Rab27a, Munc13-4-dependent manner (*Messenger et al., 2018*; *Savina et al., 2003*). These results raise the possibility that exosomes may be secreted as a byproduct of the $Ca^{2+}$-dependent plasma membrane repair process that occurs after membrane disruption in vivo. Perforation of the plasma membrane results in an influx of extracellular $Ca^{2+}$ into the cytoplasm and triggers a repair cascade that includes mobilization and fusion of lysosomes with the plasma membrane (*Andrews and Corrotte, 2018*). Given the similarity of lysosomes and MVBs, it is plausible to suggest that exosomes are secreted as a byproduct of $Ca^{2+}$-dependent plasma membrane repair. However, some reports have suggested that, unlike lysosomes, MVBs do not undergo $Ca^{2+}$-dependent exocytosis (*Jaiswal et al., 2004*; *Jaiswal et al., 2002*).

To determine if exosomes are secreted during $Ca^{2+}$-dependent plasma membrane repair, we developed an improved nanoluciferase (Nluc) reporter system to quantify exosome secretion that is sensitive, linear, and amenable to high throughput. Using this assay, we established that a $Ca^{2+}$ ionophore, a pore-forming cytolysin, and physiological mechanical stress all stimulated exosome secretion in an extracellular $Ca^{2+}$-dependent manner. We showed that annexin A6 (ANXA6), a well-characterized plasma membrane repair protein, is recruited to MVBs in the presence of $Ca^{2+}$ and demonstrated that ANXA6 is required for $Ca^{2+}$-dependent exosome secretion. We then observed that depletion of ANXA6 stalls MVBs at the periphery of cells treated with a $Ca^{2+}$ ionophore. Next, we developed a streptolysin O (SLO)-permeabilized cell reaction that recapitulates $Ca^{2+}$- and ANXA6-dependent exosome secretion. Finally, we demonstrated that the two annexin domains of ANXA6 become enriched at different membranes upon elevation of cytosolic $Ca^{2+}$. Our results demonstrate that exosome secretion is coupled to $Ca^{2+}$-dependent plasma membrane repair and that ANXA6 may serve as a potential tether for the recruitment of MVBs to the plasma membrane.

## Results

### Design and validation of an endogenous CD63-Nluc exosome secretion assay

We sought to develop a cell-based exosome secretion assay that is quantitative, sensitive, amenable to high-throughput, and able to distinguish cell debris and *bona fide* exosomes. A previous study developed a luminescence-based assay to quantify exosome secretion by inserting Nluc into the endogenous locus of the tetraspanin protein CD63 (*Hikita et al., 2018*). We obtained the HCT116 CD63Nluc-KI #17 cell line (referred to herein as CD63-Nluc cells) from this group and built upon their assay.

Sequestration of CD63 during ILV biogenesis results in the amino- and carboxyl-termini oriented to the exosome lumen. We leveraged this topology along with the relative membrane permeabilities of the substrate and an inhibitor of Nluc to develop a quantitative cellular assay to monitor the secretion of exosomes (*Figure 1A*). Furimazine, the substrate of Nluc, permeates through cellular membranes and enables luminescence from both cellular debris and intact CD63-Nluc exosomes

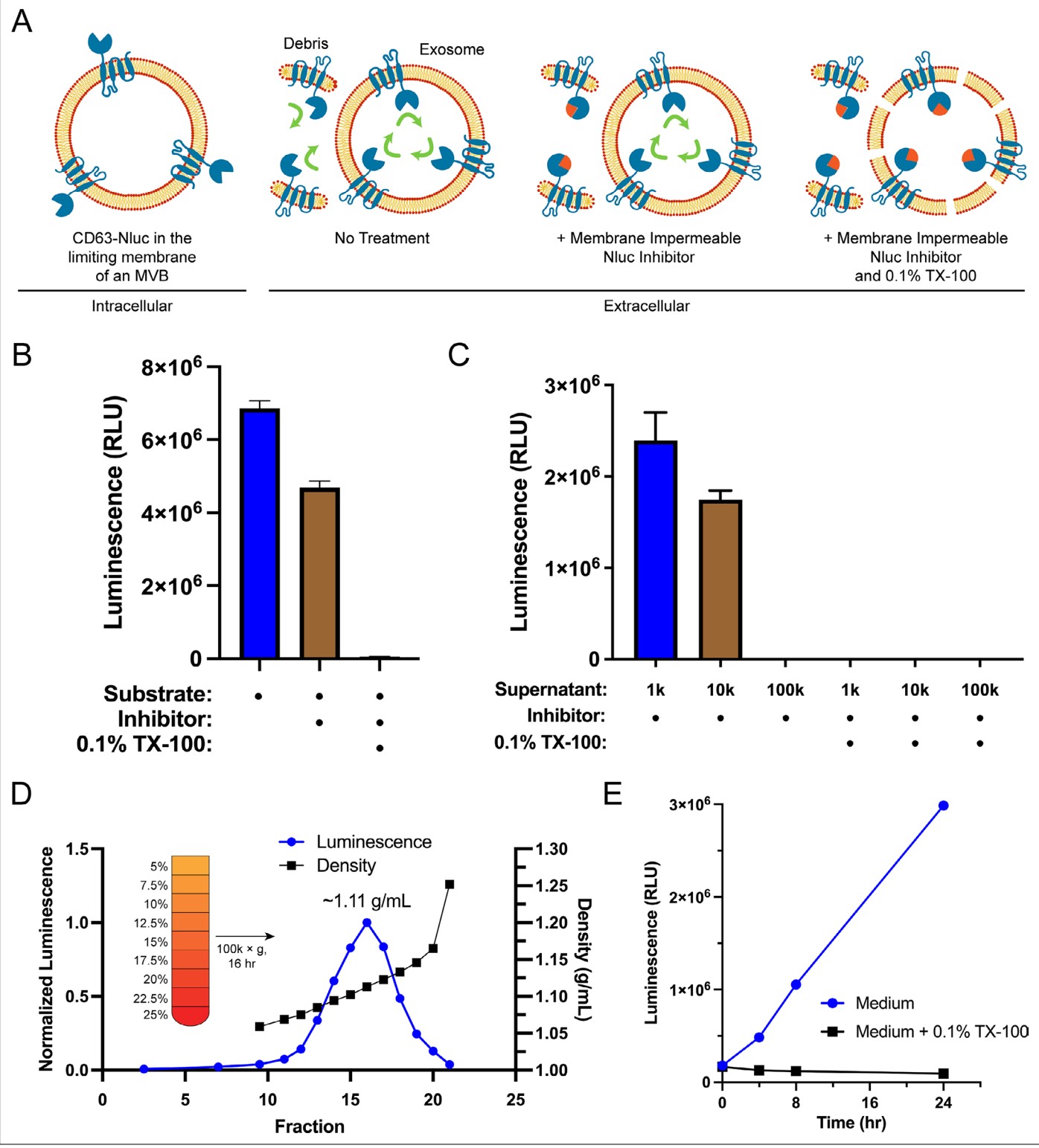

**Figure 1.** Endogenous CD63-nanoluciferase (Nluc) is a faithful reporter of exosome secretion. (**A**) Schematic illustrating the topology of CD63-Nluc in different membranes and the cellular CD63-Nluc secretion assay. (**B**) Luminescence from the conditioned medium of CD63-Nluc cells with or without furimazine, the membrane-impermeable Nluc inhibitor, and 0.1% TX-100 are shown. (**C**) Luminescence derived from the supernatant fraction of CD63-Nluc conditioned medium subjected to differential centrifugation (1k, 10k, and 100k) with or without 0.1% TX-100 are shown. (**D**) Membrane-protected luminescence (in blue circles) and buoyant density (in black squares) of CD63-Nluc conditioned medium subjected to a high-resolution linear density gradient are shown. (**E**) Membrane-protected luminescence from CD63-Nluc conditioned medium collected over 24 hr with (blue circles) or without

*Figure 1 continued on next page*

*Figure 1 continued*

0.1% TX-100 (black squares). Data plotted represent the means of three independent experiments, and error bars represent SDs. Note, for *Figure 1D and E*, the error bars are smaller than the dots in the image.

present within the conditioned medium of cultured CD63-Nluc cells. The addition of a membrane-impermeable Nluc inhibitor eliminates luminescence derived from cellular debris where Nluc is readily accessible, while not affecting the luminescence derived from intact CD63-Nluc exosomes incubated with furimazine (*Walker et al., 2017*). Finally, solubilization of cellular membranes by the addition of the non-ionic detergent Triton X-100 (TX-100) exposes all CD63-Nluc to the inhibitor. The addition of the membrane-impermeable Nluc inhibitor depleted a significant portion of luminescence derived from the conditioned medium of CD63-Nluc cells (~30%) and membrane solubilization by TX-100 reduced luminescence to background levels (*Figure 1B*).

To further validate the localization of the CD63-Nluc luminescence signal to sedimentable vesicles, we subjected the conditioned medium fraction to differential centrifugation and found that CD63-Nluc luminescence was removed after high-speed sedimentation or upon the addition of TX-100 (*Figure 1C*). Next, we obtained a partially clarified conditioned medium fraction by differential centrifugation and fractionated the material on a linear iodixanol gradient (*Figure 1D*). CD63-Nluc vesicles equilibrated at a density of 1.11 g/ml, a buoyancy similar to published reports (*Jeppesen et al., 2019*). In a time course over 24 hr, we measured secreted CD63-Nluc luminescence in the presence of Nluc inhibitor alone or with TX-100 and found the signal increased linearly, consistent with exosome secretion over time (*Figure 1E*). The luminescence signal in the presence of the Nluc inhibitor and TX-100 remained constant, consistent with cell rupture/detachment during medium change. Thus, our modified CD63-Nluc assay faithfully reported the secretion of *bona fide* CD63-positive exosomes, and we have used this method to assess the molecular requirements for exosome secretion.

## Elevation of cytosolic $Ca^{2+}$ during plasma membrane damage stimulates exosome secretion

We investigated the relationship between elevation of cytosolic $Ca^{2+}$ levels and exosome secretion. In agreement with previous reports (*Messenger et al., 2018*), influx of $Ca^{2+}$ into the cytoplasm using the $Ca^{2+}$ ionophore ionomycin (5 µM, 30 min) induced robust exosome secretion compared to the level of secretion of the vehicle control (*Figure 2A*). Exosome secretion initiated within 2 min of ionomycin treatment and was complete by 10 min (*Figure 2B*).

An influx of extracellular $Ca^{2+}$ triggers lysosome exocytosis as a means to repair plasma membrane damage (*Andrews and Corrotte, 2018*; *Corrotte et al., 2015*; *Corrotte and Castro-Gomes, 2019*; *Demonbreun and McNally, 2016*). A similar process is elicited in cells treated with a $Ca^{2+}$ ionophore (*Jaiswal et al., 2002*). We reasoned that MVBs, like lysosomes, might fuse with the plasma membrane to facilitate membrane repair. Upon treatment with the pore-forming toxin, SLO, exosomes were secreted in a $Ca^{2+}$-dependent manner (*Figure 2C*). Treatment with 400 ng/ml or 800 ng/ml of SLO significantly increased exosome production, dependent on the presence of extracellular $CaCl_2$ (1.8 mM). Similar to ionomycin, SLO induced exosome secretion 2 min after initial application and was complete after 10–20 min (*Figure 2D*). In SLO time course experiments, pore formation was synchronized by pre-incubating cells with SLO on ice (*Corrotte et al., 2015*). We noted that SLO permeabilized exosomes as well as cells. SLO treatment of conditioned medium decreased the CD63-Nluc signal when the membrane-impermeable Nluc inhibitor was present (*Figure 2—figure supplement 1A*). Thus, our assay likely underestimates SLO-induced exosome secretion. To test whether plasma membrane damage also elicits exosome secretion in a different cell line, we generated a HEK293T reporter cell line that expresses FLAG-Nluc-CD63 under expression of the weak L30 promoter. We observed an increase in FLAG-Nluc-CD63 exosome secretion from this reporter cell line in a SLO dose- and $Ca^{2+}$-dependent manner (*Figure 2—figure supplement 1B*). Application of a physiological mechanical stress also induced exosome secretion in a $Ca^{2+}$-dependent manner (*Figure 2E*). HCT116 CD63-Nluc cells were pumped slowly through a narrow-gauge needle to simulate a capillary. In the low mechanical stress regime, cells transiently experienced ~89 dyn/cm² maximum fluid shear stress, whereas in the high-mechanical stress regime, cells transiently experienced ~178 dyn/cm² maximum fluid shear stress (*Barnes et al., 2012*). This form of mechanical stress increased exosome production but only when $CaCl_2$ was present in the conditioned medium. Endothelium and circulating lymphocytes

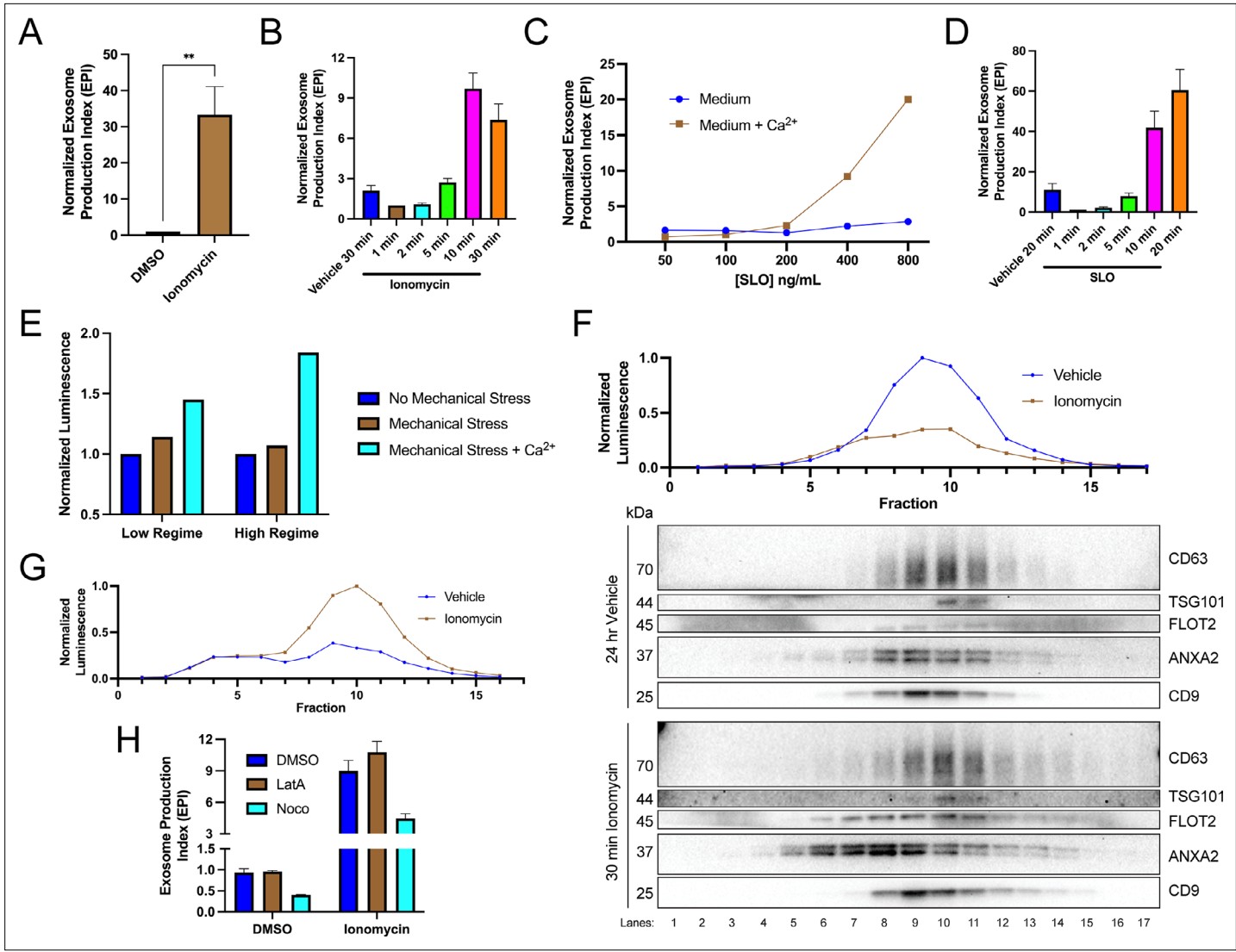

**Figure 2.** Elevation of cytosolic Ca²⁺ levels promotes exosome secretion. (**A**) Normalized exosome production from CD63-nanoluciferase (Nluc) cells treated with 5 μM ionomycin or DMSO (ionomycin vehicle) are shown. (**B**) Relative rate of exosome secretion over time is shown. CD63-Nluc cells were treated with DMSO or 5 μM ionomycin for the indicated times, and the normalized exosome production index was calculated. (**C**) Normalized exosome production from CD63-Nluc cells treated for 30 min with increasing concentrations of streptolysin O (SLO), with or without 1.8 mM extracellular Ca²⁺. Note, error bars are smaller than the dots in the image. (**D**) Relative rate of exosome secretion over time is shown. CD63-Nluc cells were treated with PBS or 250 ng/ml SLO for the indicated time points, and the normalized exosome production index was calculated. (**E**) Normalized luminescence derived from CD63-Nluc cells treated with a high or low dose of mechanical stress, with or without 1.8 mM extracellular Ca²⁺. (**F**) Iodixanol gradient fractionation of conditioned medium 100k × g pellet fraction is shown. Conditioned medium from cells treated for 30 min with 5 μM ionomycin or 24 hr vehicle is compared. Line graphs show distribution of CD63-Nluc luminescence (with membrane-impermeable inhibitor added) across the linear gradient. Immunoblots show distribution of several extracellular vesicle (EV) markers across the linear gradient. (**G**) Iodixanol gradient fractionation of conditioned medium 10k × g supernatant fractions are shown. Conditioned medium from cells treated for 4 hr with 5 μM ionomycin or 4 hr vehicle is compared. Line graphs show distribution of CD63-Nluc luminescence (with membrane-impermeable inhibitor added) across the linear gradient. (**H**) Normalized exosome production from 30 min of 5 μM ionomycin or DMSO vehicle, co-treated with DMSO vehicle, 1 μM latrunculin A (LatA), or 10 μM nocodazole (Noco) is shown. Data plotted represent the means from three independent experiments, and error bars represent SDs. Statistical significance was performed using a Student's T-test (**p<0.01).

The online version of this article includes the following source data and figure supplement(s) for figure 2:

**Source data 1.** Uncropped immunoblot images corresponding to *Figure 2*.

**Figure supplement 1.** Elevation of cytosolic Ca²⁺ levels promotes exosome secretion.

**Figure supplement 1—source data 1.** Uncropped immunoblot images corresponding to *Figure 2—figure supplement 1*.

routinely experience mechanical stress of ~95 dyn/cm$^2$ and transiently experience up to ~3000 dyn/cm$^2$ (*Barnes et al., 2012*).

Next, we used linear iodixanol density gradient fractionation to assess if the Ca$^{2+}$-dependent increase in extracellular CD63 was attributable to exosomes as opposed to the secretion of intact endosomes contained within plasma membrane-derived vesicles (*Jeppesen et al., 2019*). EVs from cells treated with ionomycin for 30 min or vehicle for 24 hr were separated on a high-resolution linear iodixanol gradient. A 30 min ionomycin treatment induced the secretion of low-buoyant density, ANXA2- and FLOT2-positive vesicles (*Figure 2F*, Fraction #8) compared to the 24 hr vehicle control. ANXA2 is a marker for a subpopulation of plasma membrane-derived EVs (*Jeppesen et al., 2019*). Alternatively, CD63 and Nluc signals equilibrated to higher buoyant density fractions corresponding to the position expected for exosomes (*Figure 2F*, Fraction #10). A comparison of ionomycin- and vehicle-treated samples showed drug treatment over equivalent times (4 hr) increased the Nluc luminescence in putative exosome fractions (*Figure 2G*, Fraction #10). Treatment of cells with ionomycin for 4 hr did not induce apoptotic cell death (*Figure 2—figure supplement 1C*; *Gil-Parrado et al., 2002*).

MVBs traffic toward the cell periphery on microtubules or actin filaments (*Mittelbrunn et al., 2015*). We considered the possibility that Ca$^{2+}$-dependent exosome secretion depended on MVB traffic on microtubules or actin filaments. Treatment with nocodazole, a microtubule polymerization inhibitor, but not latrunculin A, an actin polymerization inhibitor, reduced vehicle- and ionomycin-induced exosome secretion (*Figure 2H*, *Figure 2—figure supplement 1D and E*).

We conclude that the influx of extracellular Ca$^{2+}$ caused by several inducers of plasma membrane lesions triggers exosome secretion and that this secretion is dependent on the anterograde trafficking of MVBs on microtubules.

## Identification of candidate proteins involved in Ca$^{2+}$-dependent exosome secretion

Next, we sought to identify cytosolic proteins that are recruited in a Ca$^{2+}$-dependent manner to MVBs. Such proteins may be involved in Ca$^{2+}$-dependent exosome secretion and plasma membrane repair. CD63-Nluc cells were ruptured by homogenization, and a post-nuclear supernatant fraction was supplemented with 1 mM CaCl$_2$ or 1 mM EGTA and mixed with immobilized Nluc antibody in order to immunoprecipitate (IP) MVBs and associated peripheral membrane proteins. After washing the immunoprecipitated MVBs to remove other organelles, Ca$^{2+}$-dependent MVB binding proteins were eluted from the IP fraction using 2 mM EGTA. MVB proteins retained after the EGTA elution were solubilized using 0.2% TX-100 (*Figure 3A*). The 0.2% TX-100 elution fractions were significantly enriched in LAMP1 and diminished in GAPDH compared to the input, regardless of the presence of CaCl$_2$ or EGTA in the elution (*Figure 3B*). A gel stained for total protein showed four intense protein bands unique to the EGTA-eluted sample (*Figure 3C*). These proteins were absent when the Nluc antibody was omitted or when the post-nuclear supernatant was treated with 1 mM EGTA instead of 1 mM CaCl$_2$. Three sections of the gel were excised and analyzed by mass spectrometry (*Figure 3D*). We identified annexin A6 (ANXA6) and copine 3 (CPNE3) in the upper gel slice, annexin A2 (ANXA2) in the middle gel slice, and S100 Ca$^{2+}$-binding protein A10 (S100A10) in the lower gel slice.

## ANXA6 depletion blocks Ca$^{2+}$-dependent exosome secretion and stalls MVBs at the cell periphery

We tested the effect of knocking-down genes encoding the Ca$^{2+}$-dependent MVB-binding proteins identified in our proteomic analysis. We were unable to knock down ANXA2 using two different shRNAs. However, we were able to efficiently knock down ANXA6 and CPNE3 with two different shRNAs (*Figure 4A and B*), and in each case, Ca$^{2+}$-dependent exosome secretion decreased relative to a GFP knockdown control (*Figure 4C and D*). One of the ANXA6 shRNAs also slightly decreased constitutive exosome secretion relative to the GFP control, although the effect was relatively small (*Figure 4C*). A polyclonal knockout of ANXA6 also decreased exosome secretion relative to a non-targeting control (*Figure 4—figure supplement 1A and B*).

In order to assess how ANXA6 depletion affected exosome secretion, we visualized CD63-positive compartments after Ca$^{2+}$ influx. CD63-Nluc cells were treated with ionomycin or vehicle control for 30 min, fixed, and analyzed using immunofluorescence microscopy. We observed an accumulation

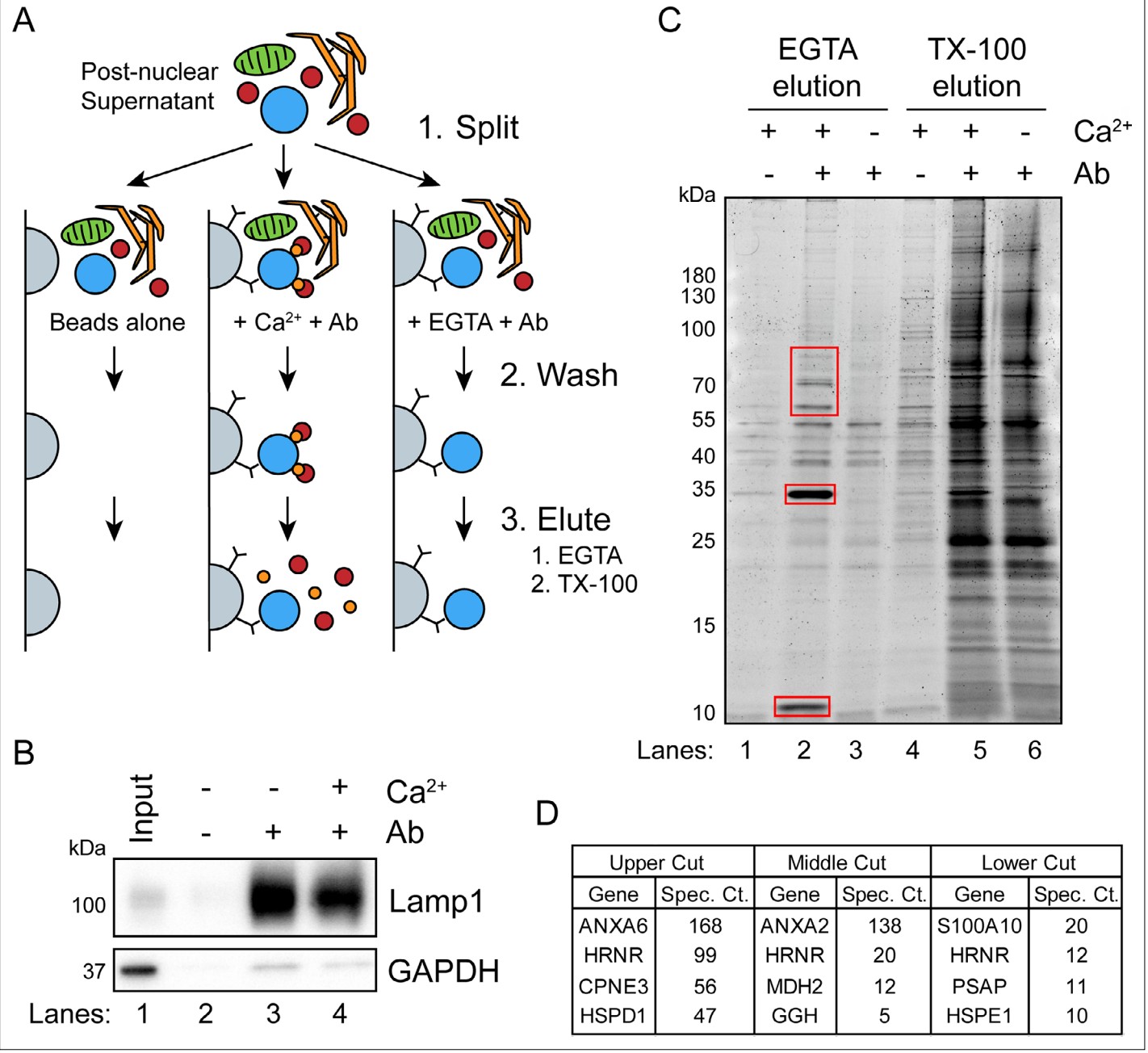

**Figure 3.** A targeted proteomics approach identifies genes important for $Ca^{2+}$-dependent exosome secretion. (**A**) Schematic illustrating the isolation of $Ca^{2+}$-dependent multivesicular body (MVB)/lysosome binding proteins (Ab: antibody; gray-beads, blue-MVB, green-mitochondria, orange-ER, red-proteins, and gold-$Ca^{2+}$). (**B**) Immunoblot analysis of LAMP1 and GAPDH from the TX-100 elutions, relative to the input, is shown. (**C**) Total protein gel (Sypro Ruby stained) of eluted fractions is shown. Red boxes indicate gel cuts sent for proteomic analysis. (**D**) Table depicting the top four proteomic hits from each gel cut are shown, excluding keratin family proteins.

The online version of this article includes the following source data for figure 3:

**Source data 1.** Uncropped immunoblot and gel images corresponding to *Figure 3*.

**Source data 2.** Mass spectrometry analysis of proteins recruited to multivesicular bodies (MVBs) in the presence of $Ca^{2+}$.

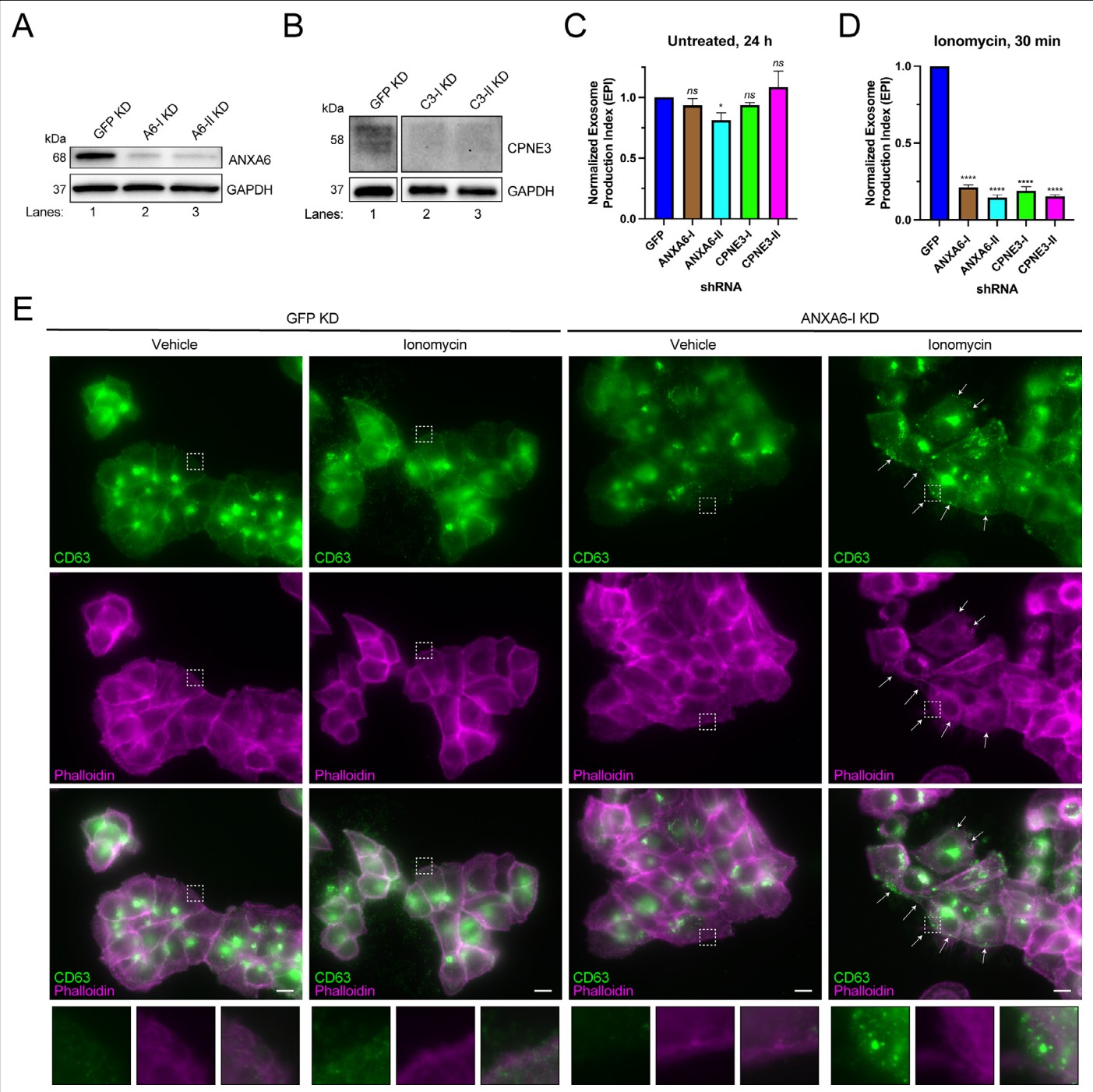

**Figure 4.** ANXA6 depletion blocks ionomycin-mediated exosome secretion and stalls multivesicular bodies (MVBs) at the cell periphery. (**A**) Immunoblot analysis of ANXA6 and GAPDH expression from GFP, ANXA6-I, and ANXA6-II shRNA CD63-nanoluciferase (Nluc) cells is shown. (**B**) Immunoblot analysis of CPNE3 and GAPDH expression from GFP, CPNE3-I, and CPNE3-II shRNA CD63-Nluc cells are shown. (**C**) Normalized exosome production derived from GFP, ANXA6-I, ANXA6-II, CPNE3-I, and CPNE3-II shRNA CD63-Nluc cells grown in conditioned medium for 24 hr are shown. (**D**) Normalized exosome production derived from GFP, ANXA6-I, ANXA6-II, CPNE3-I, and CPNE3-II shRNA CD63-Nluc cells treated with 5 µM ionomycin for 30 min are shown. Data plotted represent the means from three independent experiments, and error bars represent SDs. Statistical significance was performed using an ANOVA (*p<0.05, ****p<0.0001, and ns = not significant). (**E**) CD63 immunofluorescence and phalloidin staining of GFP or ANXA6-I shRNA CD63-Nluc cells after 30 min of DMSO or 5 µM ionomycin treatment are shown. White arrows indicate peripheral CD63 puncta. Scale bars: 10 µm.

The online version of this article includes the following source data and figure supplement(s) for figure 4:

**Source data 1.** Uncropped immunoblot images corresponding to *Figure 4*.

*Figure 4 continued on next page*

*Figure 4 continued*

**Figure supplement 1.** Exosome secretion from polyclonal ANXA6 KO cells.

**Figure supplement 1—source data 1.** Uncropped immunoblot images corresponding to *Figure 4—figure supplement 1*.

of CD63-positive vesicles at the cell periphery, particularly in ionomycin-treated ANXA6 knockdown cells (*Figure 4E*). We suggest that ANXA6 may be necessary for $Ca^{2+}$-dependent exosome secretion, possibly at the point of docking between the MVB and the plasma membrane.

## Biochemical reconstitution of $Ca^{2+}$- and ANXA6-dependent exosome secretion in permeabilized cells

Many studies probing genes that may contribute to exosome secretion have relied on depletion or overexpression experiments in live cells. We sought to develop an assay in permeabilized cells to allow a direct assessment of the roles of different gene products in exosome secretion.

SLO is a potent bacterial toxin secreted by group A streptococci that forms stable pores within cholesterol-containing biological membranes in a temperature-dependent manner (*Alouf, 1980*). SLO has been used to permeabilize cells to reconstitute various intracellular membrane trafficking and organelle exocytosis reactions because it forms stable pores that are large enough to allow the diffusion of large proteins, but not intact organelles, across the plasma membrane (*Apodaca et al., 1996*; *Funato et al., 1997*; *Martys et al., 1995*). We leveraged the cholesterol- and temperature-dependent properties of SLO to establish a permeabilized cell reaction that would allow us to study exosome secretion upon plasma membrane damage.

Reaction components including an ATP regeneration system (ATP$^r$), GTP, cytosol, and $Ca^{2+}$ were mixed with SLO-permeabilized cells and incubated at 30°C and evaluated for exosome secretion just as in the cell-based exosome secretion assay (*Figure 5A*). We observed that addition of either rat liver cytosol or $Ca^{2+}$ promoted exosome secretion slightly and that the addition of both rat liver cytosol and $Ca^{2+}$ promoted exosome secretion ~12-fold over the control condition (*Figure 5B*). The addition of HCT116 WT cytosol and $Ca^{2+}$ promoted exosome secretion ~15-fold over the control condition (*Figure 5C*). To assess the energy dependence of this reaction, we repeated the experiment using nucleotide-depleted cytosol in either the absence or presence of the ATP$^r$ and GTP (*Figure 5D*). We observed that the $Ca^{2+}$-only reaction was stimulated by the addition of the ATP$^r$ and GTP (columns 3 and 7), whereas the stimulatory effect obtained with cytosol alone appeared to be nucleotide-independent (columns 2 and 6). The $Ca^{2+}$ conditional and partial ATP dependence of the reaction may reflect distinct pools of MVBs, some perhaps already bound to the plasma membrane.

The participation of ANXA6 in the reconstitution was assessed with blocking IgG antibodies targeting epitopes on either GFP or ANXA6. Addition of an anti-GFP antibody slightly decreased exosome secretion whereas the equivalent addition of an anti-ANXA6 antibody (whose specificity was validated by immunoblot analysis in *Figure 4A* and *Figure 4—figure supplement 1A*) decreased exosome secretion to a background level seen in a reaction without added cytosol (*Figure 5E*). These results suggest a direct role of ANXA6 in the docking of MVBs at the cell surface.

## ANXA6 truncations localize to different membranes upon ionomycin treatment

After demonstrating that ANXA6 is required for $Ca^{2+}$-dependent exosome secretion in intact and permeabilized cells, we sought to probe the membrane recruitment of distinct domains of ANXA6 upon $Ca^{2+}$ influx. Previous studies have demonstrated that ANXA6 is recruited to a peripheral 'repair cap' that is formed at the site of plasma membrane lesions (*Demonbreun et al., 2016*). Unlike other members of the annexin protein family, ANXA6 contains two complete annexin domains, ANXA6(N) and ANXA6(C). To probe the roles of these domains, we generated fluorescent ANXA6 full length (FL), ANXA6(N), and ANXA6(C) fusion constructs and probed their localization in U-2 OS cells by superresolution microscopy.

We observed that ANXA6(FL) maintains a diffuse cytoplasmic distribution in unperturbed cells (*Figure 6A*). However, upon addition of ionomycin, ANXA6(FL) was recruited to both the plasma membrane (as indicated by wheat germ agglutinin [WGA] counterstain) and to intracellular vesicles, a portion of which were CD63-positive (*Figure 6A*). We also observed the budding of plasma

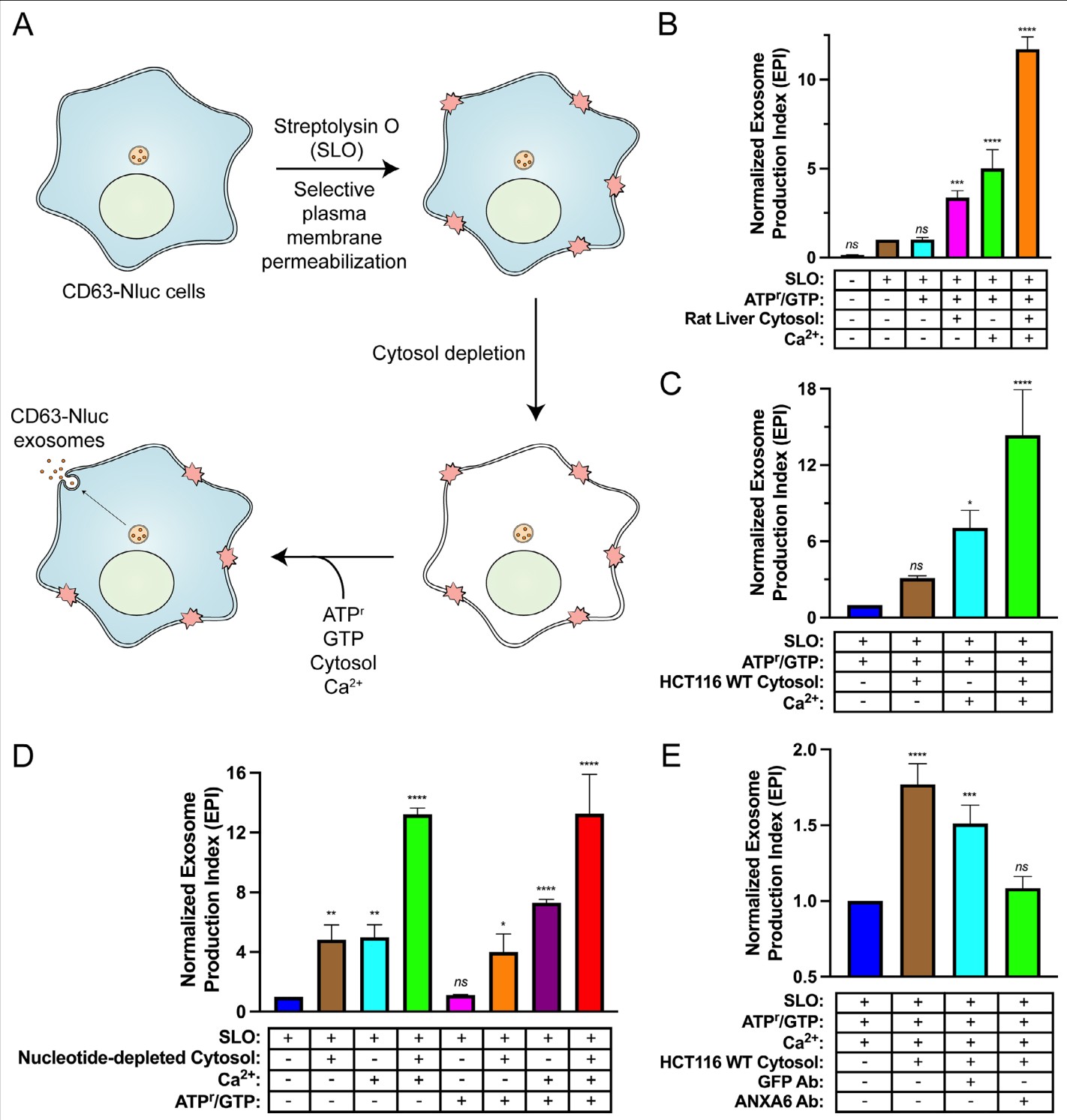

**Figure 5.** Biochemical reconstitution of $Ca^{2+}$- and ANXA6-dependent exosome secretion in permeabilized cells. (**A**) Schematic illustrating the permeabilized cell exosome secretion assay. (**B**) Permeabilized cell exosome secretion reactions with or without SLO, ATP regeneration system (ATP$^r$)/GTP, rat liver cytosol, and $Ca^{2+}$ are indicated. (**C**) Permeabilized exosome secretion assays with or without HCT116 WT cytosol and $Ca^{2+}$ are shown. (**D**) ATP requirements for the permeabilized exosome secretion assay. Reactions with or without nucleotide-depleted rat liver cytosol, $Ca^{2+}$, and ATP$^r$/GTP are indicated. (**E**) Requirement of ANXA6 in the permeabilized exosome secretion assay. Reactions with or without HCT116 WT cytosol, an anti-GFP rabbit IgG antibody, and an anti-ANXA6 rabbit IgG antibody are depicted. Data plotted represent the means from three independent experiments, and error bars represent SDs. Statistical significance was performed using an ANOVA (*p<0.05, **p<0.01, ***p<0.001, ****p<0.0001, and ns = not significant).

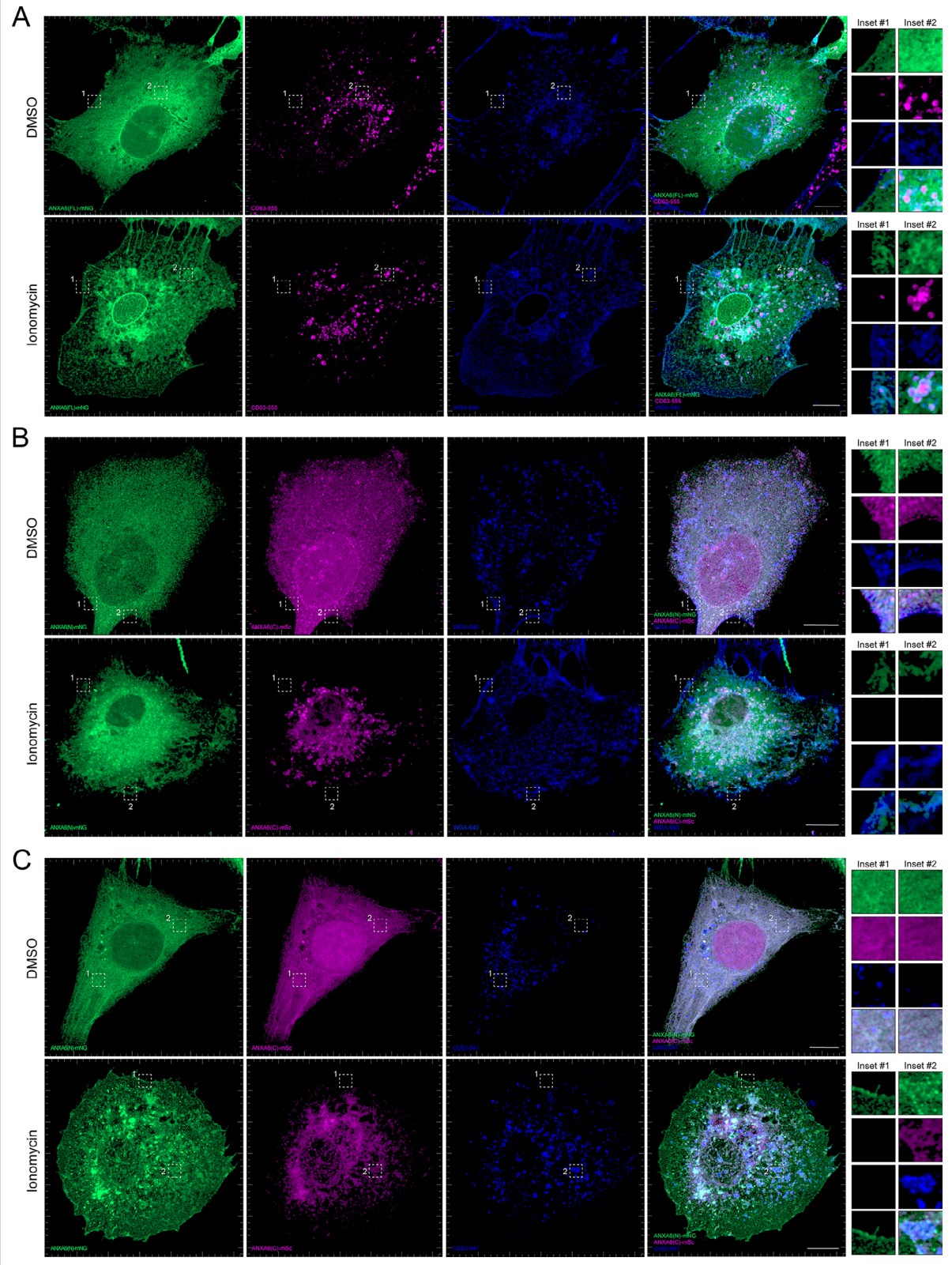

**Figure 6.** Localization of full-length and truncated ANXA6 constructs with or without ionomycin treatment. (**A**) Airyscan 3D projection of U-2 OS cells expressing ANXA6(FL)-mNG with endogenous CD63 and wheat germ agglutinin (WGA) counterstain upon treatment with DMSO or 1 µM ionomycin for 10 min. Green: ANXA6(FL)-mNG; magenta: endogenous CD63 immunofluorescence; blue: WGA CF640R conjugate. Scale bars: 10 µm. (**B**) Airyscan 3D projection of U-2 OS cells expressing ANXA6(N)-mNG and ANXA6(C)-mSc with WGA counterstain upon treatment with DMSO or 1 µM ionomycin for

*Figure 6 continued on next page*

*Figure 6 continued*

10 min. Green: ANXA6(N)-mNG; magenta: ANXA6(C)-mSc; blue: WGA CF640R conjugate. Scale bars: 10 μm. (**C**) Airyscan 3D projection of U-2 OS cells expressing ANXA6(N)-mNG and ANXA6(C)-mSc with endogenous CD63 upon treatment with DMSO or 1 μM ionomycin for 10 min. Green: ANXA6(N)-mNG; magenta: ANXA6(C)-mSc; blue: endogenous CD63 immunofluorescence. Scale bars: 10 μm.

The online version of this article includes the following video(s) for figure 6:

**Figure 6—video 1.** Rotating 3D projection of DMSO-treated U-2 OS merge in *Figure 6A*.
https://elifesciences.org/articles/86556/figures#fig6video1

**Figure 6—video 2.** Rotating 3D projection of ionomycin-treated U-2 OS merge in *Figure 6A*.
https://elifesciences.org/articles/86556/figures#fig6video2

**Figure 6—video 3.** Rotating 3D projection of DMSO-treated U-2 OS merge in *Figure 6B*.
https://elifesciences.org/articles/86556/figures#fig6video3

**Figure 6—video 4.** Rotating 3D projection of ionomycin-treated U-2 OS merge in *Figure 6B*.
https://elifesciences.org/articles/86556/figures#fig6video4

**Figure 6—video 5.** Rotating 3D projection of DMSO-treated U-2 OS merge in *Figure 6C*.
https://elifesciences.org/articles/86556/figures#fig6video5

**Figure 6—video 6.** Rotating 3D projection of ionomycin-treated U-2 OS merge in *Figure 6C*.
https://elifesciences.org/articles/86556/figures#fig6video6

membrane-derived vesicles in the presence of ionomycin. These may correlate to the low-buoyant density vesicles marked by ANXA2 and FLOT2 that were released from ionomycin-treated cells (*Figure 2F*, Fraction #8). Next, we expressed fluorescently tagged ANXA6(N) and ANXA6(C) constructs and assessed their localization upon the addition of DMSO or ionomycin, relative to the plasma membrane and CD63, respectively (*Figure 6B and C*). Similar to ANXA6(FL), ANXA6(N) and ANXA6(C) displayed cytoplasmic localization in unstimulated cells. Upon ionomycin treatment, we observed recruitment of ANXA6(N), but not ANXA6(C) to the cell periphery, relative to the WGA counterstain (*Figure 6B*). In contrast to ANXA6(N), we observed that ANXA6(C) became enriched at intracellular membrane structures, a portion of which were positive for CD63 (*Figure 6C*). These two domains may serve to dock MVBs to the plasma membrane.

## Discussion

Our results suggest that exosome secretion is coupled to $Ca^{2+}$-dependent plasma membrane repair in HCT116 and HEK293T cells (*Figure 7*). We established cellular and biochemical exosome secretion assays to recapitulate and interrogate this process. Targeted proteomics was used to demonstrate that ANXA6 is recruited to MVBs in the presence of $Ca^{2+}$. We show, both in intact and permeabilized cells, that ANXA6 is required for $Ca^{2+}$-dependent exosome secretion. We then provide evidence that that ANXA6 depletion stalls MVBs at the cell periphery upon $Ca^{2+}$ ionophore treatment. Finally, we demonstrate that truncations of ANXA6 are enriched at different cellular membranes upon cytosolic $Ca^{2+}$ elevation.

### Exosomes are secreted upon damage to the plasma membrane

We demonstrate that SLO-treated or mechanically stressed HCT116 CD63-Nluc and SLO-treated HEK293T FLAG-Nluc-CD63 cells secrete exosomes in response to $Ca^{2+}$-dependent plasma membrane repair (*Figure 2* and *Figure 2—figure supplement 1B*). Such treatments may mimic conditions of physiologic stress in vivo and lead to wound-induced EV secretion. This may contribute to the diversity of EVs present in biological fluids, including in samples used for liquid biopsy. Plasma membrane repair is common, especially in certain tissues. In rats, 6.5% of cells in the vascular endothelium of the aorta undergo plasma membrane repair at a given time (*Yu and McNeil, 1992*); 20% of muscle cells repair their membrane after muscle contractions (*McNeil and Khakee, 1992*). Motile cancer cells also have high rates of plasma membrane repair (*Bouvet et al., 2020*).

Interestingly, *Whitham, 2018* demonstrated that exercise stimulates the release of EV-associated proteins and myokines into circulating plasma (*Whitham, 2018*). In this study, a cohort of human participants had blood drawn from their femoral artery before and after 60 min of acute exercise on a cycle ergometer. Label-free quantitative proteomic analysis showed an increase in proteins such

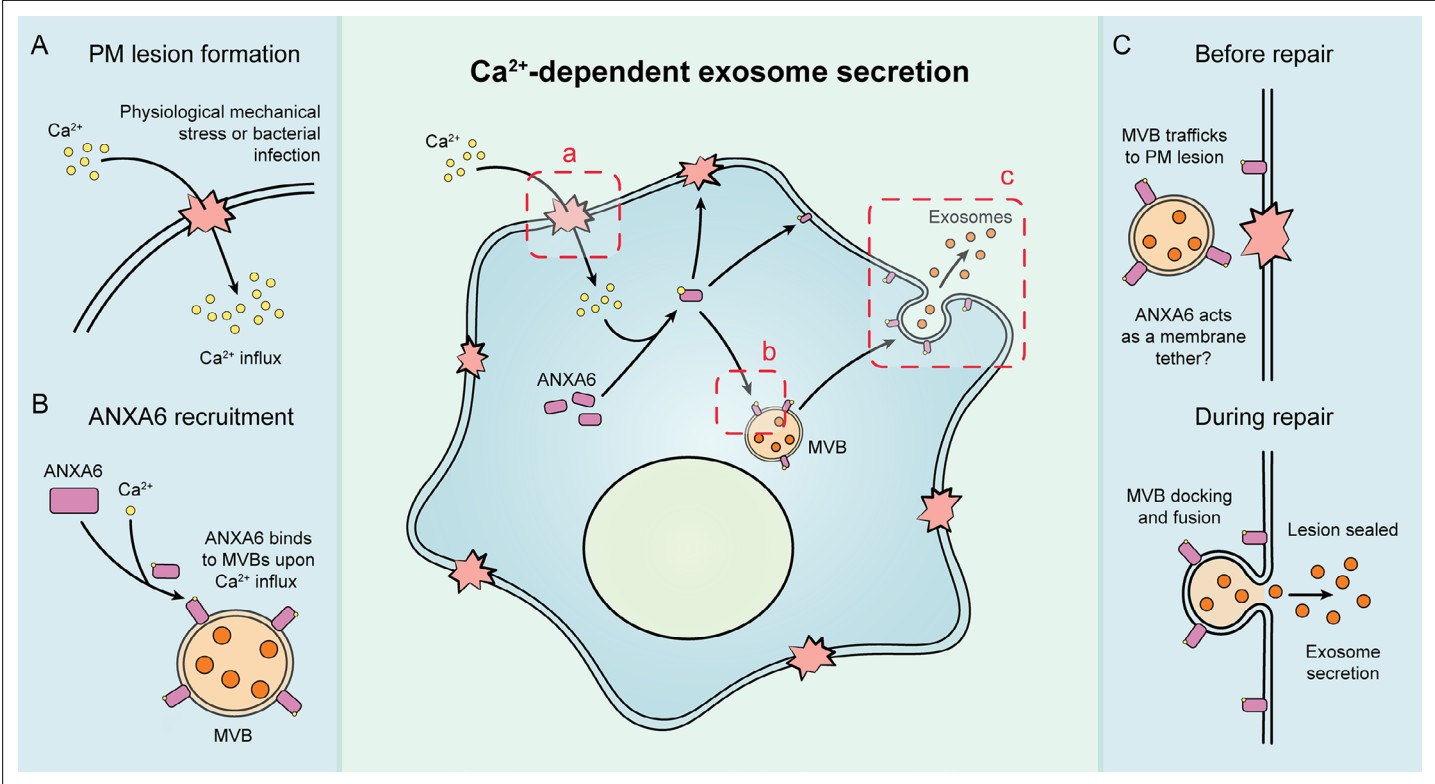

**Figure 7.** Schematic depicting the current model of $Ca^{2+}$- and ANXA6-dependent exosome secretion. (**A**) Upon physiological mechanical stress or bacterial infection, plasma membrane lesions form. This results in the flow of $Ca^{2+}$ from the extracellular space into the cytoplasm. (**B**) This influx of $Ca^{2+}$ mediates the recruitment of ANXA6 to multivesicular bodies (MVBs), which are then transported on microtubules to a plasma membrane lesion. (**C**) The MVBs then dock at the plasma membrane (with ANXA6 serving as a putative membrane tether) and undergo fusion, resulting in plasma membrane repair and exosome secretion.

as CD63, TSG101, CD9, and ANXA2 within EVs isolated from plasma by differential centrifugation postexercise. It is tempting to speculate that the increase in EV secretion is caused by increased plasma membrane repair. We postulate that exosome secretion during plasma membrane repair may allow for simple detection of tissue-specific damage during the routine clinical analysis of blood or urine samples.

The biological functions of exosomes released during $Ca^{2+}$-stimulated secretion remain unclear. Much attention has focused on the role of exosomes in intercellular communication (*Meldolesi, 2018*). A recent study from our laboratory established quantitative assays to measure the functional delivery of exosome cargo to recipient cells (*Zhang and Schekman, 2023*). Functional transfer of exosome cargo was inefficient, whereas functional intercellular transfer occurred quite effectively through the formation of open-ended membrane tubular connections. However, we do not exclude the possibility that exosomes and other EVs may functionally convey their cargo to recipient cells in specific physiological contexts (*Ridder et al., 2014*). Alternatively, a recent study showed that exosomes act as 'decoys' for pore-forming toxins such as α-toxin (*Keller et al., 2020*). During staphylococcus infection, cells upregulate exosome production. Instead of binding to the plasma membrane, α-toxin binds to exosomes. Correspondingly, we find that SLO binds exosomes at concentrations that may compete with plasma membrane binding (*Figure 2—figure supplement 1A*). $Ca^{2+}$-dependent exosome secretion allows for the rapid secretion of high levels of exosomes and may serve as a defense against bacterial toxins that perforate the plasma membrane.

Although our focus has been on exosome secretion, others have reported plasma membrane-derived vesicle secretion stimulated by plasma membrane disruptions (*Horn and Jaiswal, 2018*). Such shedded vesicles may correspond to the low-buoyant density vesicles marked by ANXA2 and FLOT2 which we find in the EV fraction produced by ionomycin treatment (*Figure 2D*).

## Do MVBs participate in plasma membrane repair?

The egress of CD63-positive exosomes from HCT116 and HEK293T cells suggests that MVBs participate in the $Ca^{2+}$-dependent repair of plasma membrane lesions. Microinjection studies in echinoderm eggs were the first to demonstrate that metazoan cells repair wounds to the plasma membrane and that this repair process is dependent upon the presence of extracellular $Ca^{2+}$ (*Chambers, 1917*; *Heilbrunn, 1930a*; *Heilbrunn, 1930b*). Since then, a variety of mechanisms by which metazoan cells repair their plasma membrane (e.g. membrane shedding, organelle exocytosis, clot formation/wound constriction, and membrane internalization) has been proposed (*Andrews and Corrotte, 2018*; *Cooper and McNeil, 2015*; *Demonbreun and McNally, 2016*; *Horn and Jaiswal, 2018*).

*Jaiswal et al., 2002* used total internal reflection fluorescence microscopy to identify the primary membrane compartment responsible for $Ca^{2+}$-dependent exocytosis in non-secretory cells. In their studies, the addition of a $Ca^{2+}$ ionophore elicited the fusion of plasma membrane-proximal lysosomes with little or no evidence of involvement of early endosomes, late endosomes/MVBs, or vesicles derived from either the endoplasmic reticulum or Golgi apparatus. Our data that CD63-positive compartments fuse at the plasma membrane upon $Ca^{2+}$ influx is consistent with their findings. However, our results also suggest that endosomes/MVBs can undergo $Ca^{2+}$-dependent exocytosis. We conclude this based on the secretion of MVB cargo vesicles (exosomes) upon $Ca^{2+}$ influx elicited by a $Ca^{2+}$ ionophore, a pore-forming toxin, or physiological levels of mechanical stress. Our results also suggest that both free and pre-docked late endosomes/MVBs participate in plasma membrane repair as exosome secretion induced by $Ca^{2+}$ ionophore treatment is partially microtubule-dependent. The markers used by *Jaiswal et al., 2002* to differentiate between late endosomes (Rab7) and lysosomes (CD63) have now been demonstrated to be present in both membrane compartments (*Humphries et al., 2011*). The combined data suggest that, in addition to lysosomes, late endosomes/MVBs can undergo $Ca^{2+}$-dependent exocytosis.

Our results are consistent with published studies conducted in sea urchin eggs and cytotoxic T lymphocytes (*Keefe et al., 2005*; *Steinhardt et al., 1994*). $Ca^{2+}$-dependent plasma membrane repair of sea urchin eggs after micropuncture with glass micropipettes has been demonstrated to be both fusion- and microtubule-dependent as suggested by sensitivity to either botulinum toxins or antibodies that block kinesin-mediated microtubule transport (*Ingold et al., 1988*; *Poulain et al., 1988*; *Schiavo et al., 1992*; *Steinhardt et al., 1994*). Endosomes pre-loaded with Alexa-488-conjugated transferrin traffic toward the plasma membrane upon treatment of cytotoxic T lymphocytes with perforin (*Keefe et al., 2005*). This result is consistent with our finding that MVBs undergo exocytosis in cells treated with SLO.

Given our findings, we propose that both MVBs and lysosomes undergo $Ca^{2+}$-dependent exocytosis during plasma membrane repair, resulting in the concurrent secretion of exosomes (*Figure 7*).

## Role of ANXA6 in exosome secretion

We find that ANXA6 is required for $Ca^{2+}$-dependent fusion of MVBs with the plasma membrane (*Figure 4*). Annexin proteins are known to be required for plasma membrane repair (*Blazek et al., 2015*; *Bouter et al., 2015*; *Boye and Nylandsted, 2016*; *Croissant et al., 2021*; *Gerke and Moss, 2002*; *Koerdt et al., 2019*). Depletion of ANXA6 compromises plasma membrane repair, and the N-terminal domain of ANXA6 is insufficient for membrane repair (*Croissant et al., 2020*; *Potez et al., 2011*; *Swaggart et al., 2014*). In addition to lysosome exocytosis, plasma membrane repair is accompanied by shedding of damaged membrane at the cell surface (*Andrews and Corrotte, 2018*). Annexins have been invoked in the capping and shedding of plasma membrane lesions (*Demonbreun et al., 2016*). Our results extend these conclusions to the fusion of MVBs at the cell surface.

Our data favors a model where ANXA6 is directly involved in tethering an MVB to the plasma membrane during exocytosis. ANXA6 is recruited to CD63-positive compartments in a $Ca^{2+}$-dependent manner (*Figure 3*). Additionally, CD63-positive membrane compartments stall near the plasma membrane in $Ca^{2+}$-stimulated, ANXA6-depleted cells (*Figure 4E*). This conclusion is strengthened by our observation that an anti-ANXA6 antibody blocks exosome secretion in permeabilized cells (*Figure 5E*). Unlike other members of the annexin family, ANXA6 contains two distinct annexin domains, possibly one for each membrane partner of a fusion pair. Upon ionomycin stimulation, we observed that the N-terminal annexin domain, but not the C-terminal annexin domain, was recruited to the plasma membrane and that the C-terminal annexin domain became enriched at intracellular

membrane structures, a portion of which were CD63-positive (*Figure 6B and C*). ANXA6 has also been demonstrated to tether liposomes in vitro in the presence of Ca$^{2+}$ (*Buzhynskyy et al., 2009*). Alternative roles in docking, such as SNARE interactions or actin remodeling as seen for other annexins, remain possible (*Gabel et al., 2015*; *Gerelsaikhan et al., 2012*). Additionally, a recent study has demonstrated that the annexin isoform ANXA11 serves as a molecular tether that allows RNA granules to 'hitchhike' on moving lysosomes for their transport (*Liao et al., 2019*).

Although we focused on the role of ANXA6 in Ca$^{2+}$-dependent exosome secretion, ANXA2 and CPNE3 were both recruited to MVBs and lysosomes in the presence of Ca$^{2+}$ and knockdown of CPNE3 inhibited MVB exocytosis (*Figures 3 and 4D*). Thus, we hypothesize that these proteins could also be involved in the docking of MVBs and lysosomes at the plasma membrane during repair.

# Materials and methods

**Key resources table**

| Reagent type (species) or resource | Designation | Source or reference | Identifiers | Additional information |
|---|---|---|---|---|
| Gene (*Homo sapiens*) | ANXA6 | Addgene | RRID: Addgene_29509 | N/A |
| Gene (*Homo sapiens*) | CPNE3 | N/A | N/A | N/A |
| Antibody | Anti-ANXA2 (rabbit monoclonal) | Abcam | Catalog #: ab178677 | WB: (1:1000) |
| Antibody | Anti-ANXA6 (rabbit monoclonal) | Abcam | Catalog #: ab201024 | WB: (1:1000), see Materials and methods for further details. |
| Antibody | Anti-CD9 (rabbit monoclonal) | Cell Signaling | RRID: AB_2798139 | WB: (1:1000) |
| Antibody | Anti-CD63 (mouse monoclonal) | BD Biosciences | RRID: AB_396297 | WB: (1:1000), IF: (1:250) |
| Antibody | Anti-CPNE3 (rabbit polyclonal) | Sigma | RRID: AB_10600703 | WB: (1:1000) |
| Antibody | Anti-Flotillin-2 (mouse monoclonal) | BD Biosciences | RRID: AB_397766 | WB: (1:1000) |
| Antibody | Anti-GAPDH (rabbit monoclonal) | Cell Signaling | RRID: AB_561053 | WB: (1:1000) |
| Antibody | Anti-GFP (rabbit polyclonal) | Torrey Pines Biolabs | RRID: AB_10013661 | See Materials and methods for further details. |
| Antibody | Anti-LAMP1 (rabbit monoclonal) | Cell Signaling | RRID: AB_2687579 | WB: (1:1000) |
| Antibody | Anti-Nluc (mouse monoclonal) | Promega | Catalog #: N7000 | See Materials and methods for further details. |
| Antibody | Anti-Tubulin (mouse monoclonal) | Abcam | RRID: AB_2241126 | IF: (1:250) |
| Antibody | Anti-TSG101 (mouse monoclonal) | GeneTex | RRID: AB_373239 | WB: (1:1000) |
| Antibody | Anti-Vinculin (rabbit monoclonal) | Abcam | RRID: AB_11144129 | WB: (1:1000) |
| Cell line (*Homo sapiens*) | HCT116 CD63Nluc-KI #17 | *Hikita et al., 2018* | N/A | Dr. Chitose Oneyama (Aichi Cancer Center Research Institute) |
| Cell Line (*Homo sapiens*) | HCT116 | Other | N/A | Cell culture facility at UC Berkeley |
| Cell line (*Homo sapiens*) | HEK293T | Other | N/A | Cell culture facility at UC Berkeley |
| Cell line (*Homo sapiens*) | HEK293T FLAG-Nluc-CD63 | This study | N/A | To assess exosome secretion in HEK293T cells |
| Cell line (*Homo sapiens*) | U-2 OS | Other | N/A | Cell culture facility at UC Berkeley |

*Continued on next page*

*Continued*

| Reagent type (species) or resource | Designation | Source or reference | Identifiers | Additional information |
|---|---|---|---|---|
| Chemical compound and drug | Ionomycin | Cayman Chemical | Catalog #: 10004974 | See Materials and Methods for further details. |
| Recombinant DNA reagent | pLKO.1_Hygro GFP shRNA | This study | N/A | shRNA target sequence: ACAACAGCCACAACGTCTAT |
| Recombinant DNA reagent | pLKO.1_Hygro ANXA2-1 shRNA | This study | N/A | shRNA target sequence: ACTTTAGAAACACGTACTTTG |
| Recombinant DNA reagent | pLKO.1_Hygro ANXA2-2 shRNA | This study | N/A | shRNA target sequence: TGAGGGTGACGTTAGCATTAC |
| Recombinant DNA reagent | pLKO.1_Hygro ANXA6-1 shRNA | This study | N/A | shRNA target sequence: AGTTGGACATGCTCGACATTC |
| Recombinant DNA reagent | pLKO.1_Hygro ANXA6-2 shRNA | This study | N/A | shRNA target sequence: CGAAGACACAATCATCGATAT |
| Recombinant DNA reagent | pLKO.1_Hygro CPNE3-1 shRNA | This study | N/A | shRNA target sequence: ACTCTATGGACCAACTAATTT |
| Recombinant DNA reagent | pLKO.1_Hygro CPNE3-2 shRNA | This study | N/A | shRNA target sequence: AGCATTCTTTCTAGGTTATTT |
| Recombinant DNA reagent | lentiCRISPR v2-Blast sgNT (non-targeting) | This study | N/A | Protospacer sequence: GCCCCGCCGCCCTCCCCTCC |
| Recombinant DNA reagent | lentiCRISPR v2-Blast sgANXA6 | This study | N/A | Protospacer sequence: AGCCTCCAGGTCCCGCTCGT |
| Recombinant DNA reagent | pLJM1-L30-FLAG-Nluc-CD63-hPGK-BlastR | This study | N/A | To express FLAG-Nluc-CD63 in various cell lines under control of the low-expression L30 promoter |
| Recombinant DNA reagent | pN1-mNeonGreen (mNG) | This study | N/A | To assess the localization of mNeonGreen-tagged proteins |
| Recombinant DNA reagent | pN1-mScarlet-i (mSc) | This study | N/A | To assess the localization of mScarlet-i-tagged proteins |
| Recombinant DNA reagent | pN1_ANXA6(FL; aa1-673)-mNG | This study | N/A | To assess localization of full-length ANXA6 |
| Recombinant DNA reagent | pN1_ANXA6(N; aa1-322)-mNG | This study | N/A | To assess localization of N-terminal truncation of ANXA6 |
| Recombinant DNA reagent | pN1_ANXA6(C; aa323-673)-mSc | This study | N/A | To assess localization of C-terminal truncation of ANXA6 |
| Software, algorithm | Prism 9 | GraphPad | RRID:SCR_002798 | N/A |
| Peptide, recombinant protein | Alexa Fluor 680 Phalloidin | Invitrogen | N/A | IF: (1:400) |
| Peptide, recombinant protein | CF640R Wheat Germ Agglutinin Conjugate | Biotium | N/A | IF: (5 µg/ml) |
| Peptide, recombinant protein | Streptolysin O (SLO) | Sigma-Aldrich | N/A | See Materials and methods for further details. |

## Cell lines, media, and general chemicals

HCT116, HCT116 CD63-Nluc, HEK293T, HEK293T FLAG-Nluc-CD63, and U-2 OS cells were cultured at 37°C in 5% $CO_2$ and maintained in Dulbecco's Modified Eagle's Medium (DMEM) supplemented with 10% fetal bovine serum (FBS) (Thermo Fisher Scientific, Waltham, MA, USA). Cells were routinely tested (negative) for mycoplasma contamination using the MycoAlert Mycoplasma Detection Kit (Lonza Biosciences). HCT116, HEK293T, and U-2 OS cells were authenticated using STR profiling at the UC Berkeley Cell Culture Facility. For the experiments detailed in *Figure 2B and C* and *Figure 2—figure supplement 1B*, we cultured HCT116 CD63-Nluc and HEK293T FLAG-Nluc-CD63 cells in $Ca^{2+}$-free DMEM (Thermo Fisher Scientific). For *Figure 2F*, HCT116 CD63-Nluc cells were incubated in EV-depleted medium. EV-depleted medium was produced by ultracentrifugation at 186,000 × *g* (40,000 RPM) for 24 hr using a Type 45Ti rotor. Ionomycin was purchased from Cayman Chemicals. Unless otherwise noted, all other chemicals were purchased from Sigma-Aldrich (St. Louis, MO, USA).

## Lentivirus production and transduction

HEK293T cells at 40% confluence within a 6-well plate were transfected with 165 ng of pMD2.G, 1.35 µg of psPAX2, and 1.5 µg of a pLKO.1-Hygro, lentiCRISPR v2-Blast, or pLJM1 plasmid using the TransIT-LT1 Transfection Reagent (Mirus Bio) as per the manufacturer's protocol. At 48 hr post-transfection, 1 ml of fresh DMEM supplemented with 10% FBS was added to each well. The lentivirus-containing medium was harvested 72 hr post-transfection by filtration through a 0.45 µm polyethersulfone (PES) filter (VWR Sciences). The filtered lentivirus was distributed in aliquots, snap-frozen in liquid nitrogen, and stored at –80°C. For lentiviral transductions, we infected HCT116 CD63-Nluc cells with filtered

lentivirus in the presence of 8 µg/ml polybrene for 24 hr, and the medium was replaced. HCT116 CD63-Nluc cells were selected using 200 µg/ml hygromycin B or 4 µg/ml blasticidin S for 8 days and 6 days, respectively. The efficiency of each gene knockdown was assessed by immunoblot analysis.

## Immunoblotting

Cells were lysed in PBS containing 1% TX-100 and a protease inhibitor cocktail (1 mM 4-aminobenzamidine dihydrochloride, 1 µg/ml antipain dihydrochloride, 1 µg/ml aprotinin, 1 µg/ml leupeptin, 1 µg/ml chymostatin, 1 mM phenylmethylsulfonyl fluoride, 50 µM N-tosyl-L-phenylalanine chloromethyl ketone, and 1 µg/ml pepstatin) and incubated on ice for 15 min. The whole cell lysate was centrifuged at $15,000 \times g$ for 10 min at 4°C and the post-nuclear supernatant was diluted with 6× Laemmli buffer (without DTT) to a 1× final concentration. Samples were heated at 95°C for 5 min and proteins resolved on 4–20% acrylamide Tris-glycine gradient gels (Life Technologies). Proteins were then transferred to polyvinylidene difluoride membranes (EMD Millipore, Darmstadt, Germany), blocked with 5% dry milk in TBS, washed 3× with TBS-T and incubated overnight with primary anti-bodies in 5% bovine serum albumin in TBS-T. The membranes were then washed again 3× with TBS-T, incubated for 1 hr at room temperature with 1:10,000 dilutions of anti-rabbit or anti-mouse secondary antibodies (GE Healthcare Life Sciences, Pittsburgh, PA, USA), washed 3× with TBS-T, washed once with TBS and then detected with ECL-2 or PicoPLUS reagents (Thermo Fisher Scientific) for proteins from cell lysates or EV isolations, respectively.

## Immunofluorescence microscopy

For immunofluorescence, we grew cells on Poly-D-Lysine (PDL)-coated coverslips which were then washed once with PBS, fixed in 4% EM-grade paraformaldehyde (PFA; Electron Microscopy Science Hatfield, PA, USA) for 15 min at room temperature, washed 3× with PBS and permeabilized/blocked in blocking buffer (2% BSA and 0.02% saponin in PBS) for 30 min at room temperature. Cells were then incubated with a 1:250 dilution of primary antibody and/or phalloidin stain overnight at 4°C, washed 3× with PBS, incubated with a 1:1000 dilution of fluorophore-conjugated secondary antibody for 1 hr at room temperature and washed 3× with PBS. Coverslips were then mounted overnight in ProLong-Gold antifade mountant with DAPI (Thermo Fisher Scientific) and sealed with clear nail polish before imaging. For microtubule visualization, cells were fixed using ice cold methanol and processed as above.

For *Figure 6*, we transfected U-2 OS cells grown on PDL-coated coverslips with the indicated plasmids using Lipofectamine 2000 as per the manufacturer's protocol. The medium containing the transfection mixture was removed ~5 hr post-transfection and replaced with DMEM supplemented with 10% FBS. Cells were treated 16 hr post-transfection with DMSO or 1 µM ionomycin with or without 5 µg/ml CF640R WGA conjugate (Biotium) as indicated for 10 min at 37°C. Treated cells were then washed with PBS, fixed in 4% EM-grade PFA for 15 min at room temperature, and processed as above. The concentrations utilized for each plasmid transfection were as follows: pN1-ANXA6(FL)-mNeonGreen (200 ng/ml), pN1-ANXA6(N)-mNeonGreen (200 ng/ml), and pN1-ANXA6(C)-mScarlet (200 ng/ml).

The images in *Figure 2—figure supplement 1* and *Figure 4* were acquired on an Echo Revolve Microscope using the 60× Apo Oil Phase, NA 1.42 objective. The images in *Figure 6* were acquired using an LSM900 confocal microscope system (ZEISS) using Airyscan 2 superresolution mode and a 63× Plan-Apochromat, NA 1.40 objective.

## EV isolation and fractionation by iodixanol buoyant density gradient equilibration

Fresh aliquots of 5, 7.5, 10, 12.5, 15, 17.5, 20, 22.5, and 25% (v/v) iodixanol solutions were prepared by mixing appropriate volumes of Solution B (0.25 M sucrose, 2 mM MgCl$_2$, 1 mM EDTA, 20 mM Tris-HCl, and pH 7.4) and Solution D (41.7 mM sucrose, 2 mM MgCl$_2$, 1 mM EDTA, 20 mM Tris-HCl, pH 7.4, and 50% (w/v) iodixanol). Iodixanol gradients were prepared by sequential 500 µl overlays of each iodixanol solution in a 5 ml SW55 tube, starting with the 25% iodixanol solution and finishing with the 5% iodixanol solution. After the addition of each iodixanol solution, the SW55 tube was flash frozen in liquid nitrogen. Complete iodixanol gradients were stored at –20°C and thawed at room temperature for 45 min prior to use.

For the iodixanol gradients detailed in *Figure 2F*, conditioned medium (240 ml) was harvested from vehicle- or ionomycin-treated HCT116 CD63-Nluc cells. All subsequent manipulations were completed at 4°C. Cells and large debris were removed by low-speed sedimentation at 1,000 × *g* for 15 min in a Sorvall R6 +centrifuge (Thermo Fisher Scientific) followed by medium-speed sedimentation at 10,000 × *g* for 15 min using a fixed angle FIBERlite F14−6×500 y rotor (Thermo Fisher Scientific). The supernatant fraction was then centrifuged at 29,500 RPM for 1.25 hr in a SW32 rotor. The high-speed pellet fractions were resuspended in PBS, pooled, loaded at the top of a prepared iodixanol gradient, and centrifuged in a SW55 rotor at 36,500 RPM for 16 hr with minimum acceleration and no brake. Fractions (200 µl) were collected from top to bottom. An aliquot of each fraction was saved for luminescence analysis, and the rest was diluted in 6× Laemmli buffer (without DTT) for immunoblot analysis. Density measurements were taken using a refractometer.

For the iodixanol gradients detailed in *Figure 2G*, we harvested conditioned medium from vehicle- or ionomycin-treated HCT116 CD63-Nluc cells grown in a 12-well plate. All subsequent manipulations were completed at 4°C. Cells and large debris were removed by low-speed sedimentation at 1000 × *g* for 15 min followed by medium-speed sedimentation at 10,000 × *g* for 15 min in an Eppendorf 5430 R centrifuge (Eppendorf, Hamburg, Germany). Aliquots (200 µl) of conditioned medium from the supernatant of the medium-speed centrifugation were loaded at the top of a prepared iodixanol gradient and centrifuged in a SW55 rotor at 36,500 RPM for 16 hr with minimum acceleration and no brake. Fractions (200 µl) were collected from top to bottom and analyzed for luminescence. Density measurements were taken using a refractometer.

## Immunoisolation of MVBs and Ca$^{2+}$-dependent binding proteins

HCT116 CD63-Nluc cells were grown to ~90% confluence in 7×150 mm dishes. Cells were scraped into 5 ml of cold PBS per plate and centrifuged at 200 × *g* for 5 min at 4°C. The cold PBS was aspirated, and the cell pellet was resuspended in two volumes of cold lysis buffer (136 mM KCl, 10 mM KH$_2$PO$_4$ pH 7.4, 1 mM DTT, protease inhibitor cocktail [see Immunoblotting section], and 6% Optiprep [w/v]). The cell slurry was passed 14 times through a 25-gauge syringe in a cold room, and the post-nuclear supernatant was prepared by centrifugation of lysed cells at 1000 × *g* for 10 min at 4°C. The post-nuclear supernatant was diluted 1:2 with lysis buffer.

Beads from 300 µl of magnetic Protein G Dynabeads (Thermo Fisher Scientific) slurry were sedimented with a magnetic tube rack and resuspended in lysis buffer. The bead slurry was split evenly into three tubes and then re-centrifuged. The diluted post-nuclear supernatant was divided between the three tubes. Tube #1 also received 1 mM CaCl$_2$ (Beads only control), tube #2 also received 1 mM CaCl$_2$ and 5 µg anti-Nluc antibody (Ca$^{2+}$-treated), and tube #3 also received 1 mM EGTA and 5 µg anti-Nluc antibody (EGTA treated). The reaction was incubated for 15 min at room temperature. Beads were washed three times with lysis buffer. A beads only control and Ca$^{2+}$-treated samples both had 2 mM CaCl$_2$ added to the wash buffer. The EGTA-treated sample was washed with lysis buffer containing 2 mM EGTA. The beads only control and Ca$^{2+}$-treated samples were eluted with 50 µl lysis buffer containing 2 mM EGTA, and the EGTA-treated sample was eluted with lysis buffer containing 2 mM CaCl$_2$. All three samples were eluted again with 50 µl lysis buffer containing 0.2% TX-100.

## CD63-Nluc exosome secretion assay

HCT116 CD63-Nluc cells were grown to ~80% confluence in 24-well plates. All subsequent manipulations were performed at 4°C. Conditioned medium (200 µl) was taken from the appropriate wells, added to a microcentrifuge tube, and centrifuged at 1000 × *g* for 15 min in an Eppendorf 5430 R centrifuge (Eppendorf, Hamburg, Germany) to remove intact cells. Supernatant fractions (150 µl) from the low-speed sedimentation were moved to a new microcentrifuge tube and centrifuged at 10,000 × *g* for 15 min to remove cellular debris. Supernatant fractions (50 µl) from this medium-speed centrifugation were then utilized to measure CD63-Nluc exosome luminescence. During these centrifugation steps, the cells were placed on ice, washed once with cold PBS, and lysed in 200 µl of PBS containing 1% TX-100 and protease inhibitor cocktail.

To measure CD63-Nluc exosome secretion, we prepared a master mix containing the membrane-permeable Nluc substrate and a membrane-impermeable Nluc inhibitor using a 1:1000 dilution of Extracellular NanoLuc Inhibitor and a 1:333 dilution of NanoBRET Nano-Glo Substrate into PBS (Promega, Madison, WI, USA). Aliquots of the Nluc substrate/inhibitor master mix (100 µl) were added

to 50 µl of the supernatant fraction from the medium-speed centrifugation and vortexed briefly, and luminescence was measured using a Promega GlowMax 20/20 Luminometer (Promega, Madison, WI, USA). An aliquot (1.5 µl) of 10% TX-100 was then added to each reaction tube for a final concentration of 0.1% TX-100, and the sample was briefly vortexed before luminescence was measured again. For the intracellular normalization measurement, the luminescence of 50 µl of cell lysate was measured using the Nano-Glo Luciferase Assay kit (Promega, Madison, WI, USA) as per the manufacturer's protocol. The exosome production index (EPI) for each sample is calculated as follows: EPI = ([medium] – [medium + 0.1% TX-100])/cell lysate.

For the CD63-Nluc exosome secretion assays in *Figure 2C and D* and *Figure 2—figure supplement 1B*, cytosol was not depleted as in the biochemical reconstitution assays detailed in *Figure 5*.

## SLO time courses

HCT116 CD63-Nluc cells were grown to ~80% confluence in 24-well PDL-coated plates. Cells were washed with 200 µl of cold PBS and incubated with 200 µl of cold (4°C) 250 ng/ml SLO in $Ca^{2+}$-free DMEM with 1 mM EGTA. The cold, SLO-containing medium was aspirated, and 200 µl of pre-warmed (37°C) $Ca^{2+}$-free DMEM with 1.8 mM $CaCl_2$ was added. Cells were incubated at 37°C for the indicated times after which exosome secretion was measured as described in paragraph 2 of 'CD63-Nluc exosome secretion assay'.

## Mechanical stress experiments

Two 15 cm plates of HCT116 CD63-Nluc cells were harvested with 10 ml of Accutase and diluted with 40 ml of $Ca^{2+}$-free DMEM. A cell slurry (16 ml) was added to three tubes and centrifuged at 300 × *g* for 5 min at room temperature. Cells were gently resuspended in either 0.5 ml $Ca^{2+}$-free DMEM or $Ca^{2+}$-free DMEM +2 mM $Ca^{2+}$ (final). Cells were pumped through a 30-gauge needle at a flow rate of 3.5 µl/s ($\tau=4Q \eta /\pi R^3$ = ~89 dyn/cm$^2$, where Q=0.0035 cm$^3$/s, $\eta$=0.01 dyn*s/cm$^2$, 30 G average internal radius = 7.94×10$^{-3}$ cm) or twice the flow rate of 7 µl/s ($\tau=4Q \eta /\pi R^3$=178 dyn/cm$^2$, where Q=0.0035 cm$^3$/s, $\eta$=0.01 dyn*s/cm$^2$, 30 G average internal radius = 7.94×10$^{-3}$ cm) using a Harvard Apparatus syringe pump (Catalog No. 98–4730). Cells were incubated for 5 min at 37°C before being placed back on ice. The cell suspension was centrifuged at 300 × *g* for 5 min, and the supernatant fraction was then filtered through a 0.45 µm PES filter. Exosome secretion was measured as described in paragraph 2 of 'CD63-Nluc exosome secretion assay,' in this assay without a cell lysate measurement.

## Isolation of cytosol from cultured human cells

HCT116 WT cells were grown to ~90% confluence in 20×150 mm dishes. All subsequent manipulations were performed at 4°C. Each 150 mm dish was washed once with 10 ml of cold PBS and then harvested by scraping into 5 ml of cold PBS. The 5 ml cell suspension was then used to harvest cells from four additional 150 mm dishes, and this process was repeated until all the cells were harvested. Cells were then collected by centrifugation at 200 × *g* for 5 min, and the supernatant fraction was discarded. The cell pellet was resuspended in 3 ml of cold hypotonic lysis buffer (20 mM HEPES, pH 7.4, 10 mM KCl, 1 mM EGTA, 1 mM DTT, and protease inhibitor cocktail) and placed on ice. After 15 min, the cell suspension was transferred to a pre-chilled 7 ml Dounce homogenizer, and the cells were mechanically lysed by ~80 strokes with a tight-fitting Dounce pestle. The lysed cells were centrifuged at 1000 × *g* for 15 min to sediment intact cells and nuclei, and the post-nuclear supernatant was then centrifuged at 32,500 RPM (~128,000 × *g*) for 30 min in an Optima XE-90 ultracentrifuge (Beckman Coulter). The supernatant (cytosol fraction) was collected conservatively without disturbing the pellet and then concentrated using a 4 ml Amicon–3 k concentrator to a final protein concentration of ~40 mg/ml. The cytosol was distributed in aliquots, snap-frozen in liquid nitrogen, and stored at –80°C until use. The rat liver cytosol was prepared as described in *Tang et al., 2020*.

## Reconstitution of CD63-Nluc exosome secretion in permeabilized cells

SLO was pre-activated in PBS containing 10 mM DTT at 37°C for 2 hr, distributed in aliquots into low-retention microcentrifuge tubes, snap-frozen in liquid nitrogen, and stored at –80°C until use. The protein concentration of each SLO batch was determined by a Bradford assay.

HCT116 CD63-Nluc cells were grown to ~80% confluence in 24-well PDL-coated plates. The PDL plate was placed on ice, and the cells were washed once with PBS containing 1 mM EGTA. The PBS

wash was aspirated, replaced with 200 µl of cold transport buffer (20 mM HEPES, pH 7.4, 250 mM D-sorbitol, 120 mM KCl, 10 mM NaCl, 2 mM MgCl$_2$, 1.2 mM KH$_2$PO$_4$, 1 mM EGTA, and protease inhibitor cocktail), supplemented with 0.6 µg/ml of pre-activated SLO, and incubated at 4°C for 15 min. Unbound SLO was aspirated, and the cells were washed once with cold transport buffer. Cells were then permeabilized by the addition of pre-warmed transport buffer containing 2 mM DTT followed by a 10 min incubation at 37°C. Permeabilized CD63-Nluc cells were then washed at 4°C in transport buffer, high-salt transport buffer (containing 1 M KOAc) and finally transport buffer (10 min each wash) to deplete cytosol.

Complete permeabilized cell exosome secretion assays (200 µl) consisted of permeabilized CD63-Nluc cells, cytosol (4 mg/ml final concentration), 20 µl 10× ATP$^r$ (10 mM ATP, 400 mM creatine phosphate, 2 mg/ml creatine phosphokinase, 20 mM HEPES, pH 7.2, 250 mM D-sorbitol, 150 mM KOAc, and 5 mM MgOAc), 3 µl of 10 mM GTP, 4 µl of 100 mM CaCl$_2$ (2 mM final concentration), and cold transport buffer. The concentration of blocking IgG antibodies utilized in *Figure 5E* was 25 µg/ml. The assembled reaction mixes were added to the permeabilized CD63-Nluc cells for 5 min on ice prior to placing the entire 24-well PDL coated plate in a 30°C water bath for 2 min to stimulate exosome secretion. Permeabilized cells were then placed back on ice, and 100 µl of each reaction supernatant was loaded into a 0.4 µm AcroPrep filter plate (Pall Corporation) and centrifuged at 1500 × *g* for 1 min in an Eppendorf 5810 R centrifuge (Eppendorf, Hamburg, Germany) to collect CD63-Nluc exosomes. During this centrifugation step, 100 µl of cold transport buffer containing 2% TX-100 and protease inhibitor cocktail was added to each well of the 24-well PDL coated plate to lyse the cells and bring the volume up to 200 µl. Filtrate aliquots (50 µl) and 50 µl of the cell lysate were used to measure exosome secretion and normalized to the number of cells per reaction, respectively, using the cell-based CD63-Nluc exosome secretion assay protocol detailed above.

For the permeabilized cell exosome secretion assay detailed in *Figure 5D*, the nucleotide-depleted rat liver cytosol was generated by using a HiTrap Sephadex G-25 Desalting Column (Cytiva Life Sciences) as per the manufacturer's protocol.

## Acknowledgements

We dedicate this work to Bob Lesch, our lab manager for the past several decades who was tragically taken from us by an accident in 2021. We thank Criss Hartzell for reading and providing helpful comments on this manuscript. We would also like to thank the staff at the UC Berkeley shared facilities, the Cell Culture Facility (Alison Killilea), the Vincent J Coates Proteomics Facility (Lori Kohlstaedt), and the DNA Sequencing Facility. JMN is supported by a National Science Foundation Graduate Research Fellowship. RS is an Investigator of the Howard Hughes Medical Institute, a Senior Fellow of the UC Berkeley Miller Institute of Science, and Scientific Director of Aligning Science Across Parkinson's Disease (ASAP).

## Additional information

### Competing interests

Randy Schekman: Founding Editor-in-Chief, eLife. The other authors declare that no competing interests exist.

### Funding

| Funder | Grant reference number | Author |
| --- | --- | --- |
| Howard Hughes Medical Institute | | Randy Schekman |
| Sergey Brin Family Foundation | | Randy Schekman |

The funders had no role in study design, data collection and interpretation, or the decision to submit the work for publication.

## Author contributions
Justin Krish Williams, Jordan Matthew Ngo, Conceptualization, Formal analysis, Investigation, Methodology, Writing - original draft, Writing – review and editing; Isabelle Madeline Lehman, Formal analysis, Investigation; Randy Schekman, Conceptualization, Formal analysis, Supervision, Funding acquisition, Writing – review and editing

## Author ORCIDs
Justin Krish Williams ⓘ http://orcid.org/0000-0002-9447-5554
Jordan Matthew Ngo ⓘ http://orcid.org/0000-0002-6566-3919
Isabelle Madeline Lehman ⓘ http://orcid.org/0009-0008-8667-401X
Randy Schekman ⓘ http://orcid.org/0000-0001-8615-6409

## Decision letter and Author response
Decision letter https://doi.org/10.7554/eLife.86556.sa1
Author response https://doi.org/10.7554/eLife.86556.sa2

---

## Additional files

### Supplementary files
• MDAR checklist

### Data availability
All data generated or analyzed during this study are included in the manuscript and supporting source data files. Source Data files have been provided for Figure 2, Figure 2 - Figure supplement 1, Figure 3, Figure 4, and Figure 4 - Figure supplement 1. Video Files have been provided for Figure 6 - Videos 1- 6.

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
