## [Editor Report]

This compelling study brings together two earlier observations: that Ca^2+^ influx can trigger exosome release from multivesicular bodies, and that plasma membrane repair after wounding requires Ca^2+^ and involves Ca^2+^-binding annexin proteins. This important work takes these earlier findings in an interesting new direction by showing that exosome release from MVBs is also triggered by Ca^2+^ influx during plasma membrane wounding and requires the annexin isoform ANX6. The study raises the interesting possibility that cell injury and repair may contribute to the release of exosomes into biological fluids.

---

## [Decision Letter]

**Decision letter after peer review:**

Thank you for submitting your article "Annexin A6 mediates calcium-dependent secretion of exosomes" for consideration by *eLife*. Your article has been favorably reviewed by 3 peer reviewers, and the evaluation has been overseen by Suzanne Pfeffer as the Senior Editor. The reviewers have opted to remain anonymous.

The reviewers have discussed their reviews with one another, and the Senior Editor has drafted this to help you prepare a revised submission. You will see from the comments that the reviewers are favorable and they would like to see the story presented in *eLife*. It is not necessary to address all the comments but some are essential, for example: please provide additional data on the kinetics of the process after ionomycin and please include additional compartment-specific markers in the immunofluorescence experiments. We leave it to you to decide which other points you wish to address.

*Reviewer #1 (Recommendations for the authors):*

Experimental request:

Lines 24, 78, 267, 381. The authors report that N- and C-terminal domains of Annexin A6 localize to "different membranes". Please use organelle/compartment-specific markers to determine the identity of at least some of the structures. Is the different localization due to one domain preferring PS more than the other? Where is the CD63 construct in these cells in relation to the two constructs? What does it look like upon SLO addition?

Also, it would be useful to the field if the authors could report, from IF data, the fraction of total MVBs that bind annexin A6 before and after ionomycin – this could lead to the search for a molecular distinction explaining why all MVBs are not capable of plasma membrane exocytosis. (For the before ionomycin condition, it may be possible to briefly permeabilize cells with digitonin to leach out cytoplasm to enable detection of any membrane-bound annexin).

*Reviewer #2 (Recommendations for the authors):*

1) Exosome release triggered by ionomycin was characterized as "rapid and robust", but this cannot be concluded with such long 30 min time points. For example, lysosome exocytosis triggered by Ca^2+^ influx is only linearly sustained for up to 10 min. The very nice permeabilized-cell reconstitution assay indicates that the exosome detection assay is sensitive enough to detect exocytosis in 2 min. This is much more consistent with the time course of PM repair (completed in 1-2 min in most cells) and, importantly, would strengthen the data by reducing concerns with ionomycin cytotoxicity.

So my suggestion is that the first part of Figure 2 includes a time course of exosome release after ionomycin with several shorter time points. If 30 min is in the linear range the results would be strengthened. If not, the main assays should be repeated. An ionomycin dose-dependent curve would also help to find the lowest effective concentration and reduce concerns about cytotoxicity.

2) As mentioned in the public review, most studies implicating ANX6 in PM repair were performed using a non-physiological form of PM wounding, using lasers. Lasers, although convenient and thus extensively used in PM repair assays, can cause drastic localized temperature increases with associated membrane protein denaturation, potentially leading to misleading results. So it would be important (and easy) to perform simple PM repair essays (propidium iodide exclusion, for example) in the ANX6-depleted cells used in this study, to confirm that ANX6 is an essential player in PM repair in this system.

3) The imaging data in Figure 4E is not strong enough to support the conclusion that MVBs are stalled at the periphery, since CD63 is also present in lysosomes. TIRF-based live imaging of exocytosis showing sudden release and diffusion into the supernatant of CD63-containing material (a behavior that should not occur with ANX6), in control but not ANX6-depleted cells, would significantly strengthen the conclusions – and also allow quantification by counting exocytosis events.

4) The imaging data in Figure 5 is also not convincing, lacking side-by-side comparisons of the same imaging mode and a PM-specific marker. The tilting images shown do not add much information – maybe rotating movies of 3D reconstructions would be more informative. Part C of Figure 5 does show an apparently different localization for each truncated ANX6 construct, but the authors seem to use this as evidence for PM tethering, while none of the individual images suggest a PM localization. Again, a PM marker would be essential here, as some form of quantification.

5) The levels of exosome release after mechanical wounding seem to be lower than with SLO – although this is hard to assess since results are expressed as an exosome production index. This could be an interesting avenue to explore since it has been suggested that PM lesions caused by pore-forming toxins or by mechanical wounds are repaired by different mechanisms, perhaps involving different organelles. Also, it could be argued that mechanical injury is more important because it occurs more frequently in vivo. Simple assays for propidium iodide or Ca^2+^ influx (to compare the extent of cell permeabilization) and for lysosomal enzyme release after each form of wounding might add interesting information here.

6) The Methods section does not provide enough information on several procedures, such as how the knockdowns were performed or how the inhibitory capacity of the anti-ANX6 antibody used in the permeabilized cell assay was validated (the table seems to suggest this was the same as used in westerns, but there is no comment on the 'knockdown validation' mentioned in the text).

7) It is not clear why the Lamp1 IF (Figure 2 Figure supplement E) was done – the results are just mentioned but not discussed. This same condition (ionomycin) was shown decades ago to trigger massive exocytosis of lysosomes, detected as punctate PM-associated Lamp1. So the peripheral punctate signal seen is likely to include Lamp1 transferred to the PM, and not only a peripheral displacement as mentioned.

*Reviewer #3 (Recommendations for the authors):*

In the manuscript "Annexin A6 mediates calcium-dependent secretion of exosomes," Williams et al., report that secretion of endosome-derived exosomes is enhanced by a calcium-dependent response to damage to the plasma membrane of cells. The authors present convincing evidence that in response to the influx of calcium that follows damage to the plasma membrane annexin A6 is recruited to multivesicular bodies (MVBs) and likely serves to tether the MVBs to the plasma membrane causing a concomitant release of exosomes.

1. In the first sentence of the Discussion, the authors specify that their results suggest that exosome secretion is coupled to calcium-dependent plasma membrane repair in HCT116 cells and indeed they present strong evidence that this is the case. Although the authors did use U2-OS cells to investigate the role of annexin A6 domains, it is not clear if this process is restricted to HCT116 cancer cells or a more general phenomenon for both normal and neoplastic cells. This reviewer is not suggesting that every experiment needs to be performed in multiple cell lines but rather that a few key experiments be repeated in at least one other cell line, perhaps normal epithelial cells or fibroblasts. This would strengthen the manuscript.

2. The authors state (sentence 262, page 10) that "…ANXA6(FL) was localized to the plasma membrane and to intracellular vesicles (Figure 6B)" and also note (sentence 268, page 11) that "…ANXA6(C) was observed on intracellular vesicles". What are these intracellular vesicles? Are they referring to intracellular compartments such as MVBs or are they referring to the intraluminal vesicles inside MVBs that are destined to be released as exosomes? From the model (Figure 7), it is suggested that annexin A6 binds to MVBs so presumably, the implication from Figure 6 is that annexin A6 localizes to MVBs following ionomycin treatment. But that is not demonstrated. It would improve the quality of the data in Figure 6 if cells labeled for both annexin A6 and CD63 (or other MVB marker) were imaged to see the extent of co-association with MVB structures.

3. Please clarify if annexin A6 itself is present in/on exosomes themselves or is absent from these.

4. Related to the question above, as annexin A6 can localize to the plasma membrane in ionomycin-treated cells, this would imply that microvesicles that bud off from the plasma membrane may also contain annexin A6. Thus, it is possible that annexin A6-positive microvesicles are released after plasma membrane damage or perturbation of the intracellular calcium level. The authors should comment on this possibility. The authors state that in response to ionomycin treatment, they observe the budding of plasma membrane-derived vesicles. If annexin A6 is present outside the cell, where does it partition after iodixanol density gradient fractionation – with CD63 or with annexin A2 and FLOT2?

5. Also related to the questions above, the authors use their endogenous CD63-Nluc system to convincingly demonstrate that specifically exosome secretion is enhanced as a consequence of plasma membrane repair but what about other types of extracellular vesicles such as plasma membrane-derived microvesicles? Is the release of microvesicles also enhanced by the plasma membrane repair process?

6. Although not directly addressed in the Discussion, this reviewer is left with the impression that the authors are hinting that exosome secretion is more a byproduct of plasma membrane repair rather than a means of intercellular communication. In other words, the cell needs the membrane material from the MVB to patch and repair holes in the plasma membrane and exosome ejection from the cell is a secondary (perhaps even irrelevant) consequence. Obviously, these two possibilities are not mutually exclusive. Nevertheless, the authors are encouraged to speculate about which possibility they favor and how their findings might change our understanding of the cell biology of exosome secretion.

7. The authors might consider placing their findings in the context of Annexin A11 acting as a tether for the transport of RNA granules (Y-C Liao et al., Cell 179:147-164, 2019).

---

## [Author Response]

Reviewer #1 (Recommendations for the authors):Experimental request:Lines 24, 78, 267, 381. The authors report that N- and C-terminal domains of Annexin A6 localize to "different membranes". Please use organelle/compartment-specific markers to determine the identity of at least some of the structures. Is the different localization due to one domain preferring PS more than the other? Where is the CD63 construct in these cells in relation to the two constructs? What does it look like upon SLO addition?

We thank the reviewer for their suggestion. We have conducted additional imaging experiments to include markers for MVBs (endogenous CD63 immunofluorescence) and the plasma membrane (wheat germ agglutinin counterstain). These results are now presented in the new Figure 6. Rotating movies of all of the 3D reconstructions are included as Figure 6 – Videos 1 through 6.

Also, it would be useful to the field if the authors could report, from IF data, the fraction of total MVBs that bind annexin A6 before and after ionomycin – this could lead to the search for a molecular distinction explaining why all MVBs are not capable of plasma membrane exocytosis. (For the before ionomycin condition, it may be possible to briefly permeabilize cells with digitonin to leach out cytoplasm to enable detection of any membrane-bound annexin).

We are also keenly interested in a molecular description as to why only a subset of MVBs are capable of plasma membrane exocytosis. Unfortunately, we are unable to conclude the proportion of ANXA6 that is bound to membranes before and after ionomycin treatment due to likely artifacts. In particular, if we were to permeabilize cells with digitonin in medium supplemented with Ca^2+^, we believe annexin proteins would most likely be relocalized to membranes. If permeabilization were conducted in medium without Ca^2+^, annexin proteins that were previously bound to membrane would likely be solubilized.

Reviewer #2 (Recommendations for the authors):1) Exosome release triggered by ionomycin was characterized as "rapid and robust", but this cannot be concluded with such long 30 min time points. For example, lysosome exocytosis triggered by Ca^2+^ influx is only linearly sustained for up to 10 min. The very nice permeabilized-cell reconstitution assay indicates that the exosome detection assay is sensitive enough to detect exocytosis in 2 min. This is much more consistent with the time course of PM repair (completed in 1-2 min in most cells) and, importantly, would strengthen the data by reducing concerns with ionomycin cytotoxicity.So my suggestion is that the first part of Figure 2 includes a time course of exosome release after ionomycin with several shorter time points. If 30 min is in the linear range the results would be strengthened. If not, the main assays should be repeated. An ionomycin dose-dependent curve would also help to find the lowest effective concentration and reduce concerns about cytotoxicity.

We thank the reviewer for their experimental suggestion. We have conducted short time course experiments for both ionomycin and SLO treatment as described in our response to Reviewer #1 (with time points at 1, 2, 5, 10, and 20 or 30 min) and these experimental results are included as the new Figure 2B and 2D.

For most of our cell assays, we chose to measure exosome concentration at the completion of a reaction where the concentration is no longer time-dependent. Processing many samples makes it technically difficult to consistently measure exosome secretion in the linear range.

Regarding the reviewer’s concerns about ionomycin toxicity, we conducted all of our experiments in DMEM supplemented with 10% FBS. Ionomycin and other ionophores bind to BSA in FBS, thus the concentration of available ionomycin is likely much lower than the amount was added to the incubation (DOI: 10.1016/B978-0-12-374841-6.00005-0). We hope that this information, in conjunction with both the short ionomycin and SLO time course experiments, allay the reviewer’s concerns.

2) As mentioned in the public review, most studies implicating ANX6 in PM repair were performed using a non-physiological form of PM wounding, using lasers. Lasers, although convenient and thus extensively used in PM repair assays, can cause drastic localized temperature increases with associated membrane protein denaturation, potentially leading to misleading results. So it would be important (and easy) to perform simple PM repair essays (propidium iodide exclusion, for example) in the ANX6-depleted cells used in this study, to confirm that ANX6 is an essential player in PM repair in this system.

We agree that laser ablation can induce artifacts and is an imperfect way to study plasma membrane repair. However, it has also been demonstrated that ANXA6 is required for the repair of SLO lesions in cultured cells (DOI: 10.1074/jbc.M110.187625). We added a reference to this publication in the Discussion section.

3) The imaging data in Figure 4E is not strong enough to support the conclusion that MVBs are stalled at the periphery, since CD63 is also present in lysosomes. TIRF-based live imaging of exocytosis showing sudden release and diffusion into the supernatant of CD63-containing material (a behavior that should not occur with ANX6), in control but not ANX6-depleted cells, would significantly strengthen the conclusions – and also allow quantification by counting exocytosis events.

We thank the reviewer for their suggestion. However, we recently attended a research conference and interacted with another group that has conducted these TIRF experiments. Because these authors are preparing to write a manuscript detailing their findings, we have not conducted these TIRF experiments at this time.

4) The imaging data in Figure 5 is also not convincing, lacking side-by-side comparisons of the same imaging mode and a PM-specific marker. The tilting images shown do not add much information – maybe rotating movies of 3D reconstructions would be more informative. Part C of Figure 5 does show an apparently different localization for each truncated ANX6 construct, but the authors seem to use this as evidence for PM tethering, while none of the individual images suggest a PM localization. Again, a PM marker would be essential here, as some form of quantification.

We thank the reviewer for their suggestion. We conducted additional imaging experiments to include markers for MVBs (endogenous CD63 immunofluorescence) and the plasma membrane (wheat germ agglutinin counterstain) and have presented these data as the new Figure 6. We have included rotating movies of all of the 3D reconstructions as Figure 6 – Videos 1 through 6.

5) The levels of exosome release after mechanical wounding seem to be lower than with SLO – although this is hard to assess since results are expressed as an exosome production index. This could be an interesting avenue to explore since it has been suggested that PM lesions caused by pore-forming toxins or by mechanical wounds are repaired by different mechanisms, perhaps involving different organelles. Also, it could be argued that mechanical injury is more important because it occurs more frequently in vivo. Simple assays for propidium iodide or Ca^2+^ influx (to compare the extent of cell permeabilization) and for lysosomal enzyme release after each form of wounding might add interesting information here.

While performing our SLO and mechanical wounding experiments, we routinely took trypan blue exclusion readings pre- and post-treatment. We always observed more trypan blue staining posttreatment in the SLO-treated cells compared to the mechanically wounded cells. This observation indicates that the increased exosome secretion in SLO-treated cells compared to the mechanically wounded cells is at least in part the result of higher cell permeabilization.

6) The Methods section does not provide enough information on several procedures, such as how the knockdowns were performed or how the inhibitory capacity of the anti-ANX6 antibody used in the permeabilized cell assay was validated (the table seems to suggest this was the same as used in westerns, but there is no comment on the 'knockdown validation' mentioned in the text).

We have clarified in the text that the specificity of the anti-ANXA6 antibody utilized in the permeabilized-cell reconstitution assay was validated by immunoblot analysis in both Figure 4A and Figure 4 —figure supplement 1A.

7) It is not clear why the Lamp1 IF (Figure 2 Figure supplement E) was done – the results are just mentioned but not discussed. This same condition (ionomycin) was shown decades ago to trigger massive exocytosis of lysosomes, detected as punctate PM-associated Lamp1. So the peripheral punctate signal seen is likely to include Lamp1 transferred to the PM, and not only a peripheral displacement as mentioned.

We thank the reviewer for this comment and have removed the original Figure 2 —figure supplement E from the manuscript.

Reviewer #3 (Recommendations for the authors):In the manuscript "Annexin A6 mediates calcium-dependent secretion of exosomes," Williams et al., report that secretion of endosome-derived exosomes is enhanced by a calcium-dependent response to damage to the plasma membrane of cells. The authors present convincing evidence that in response to the influx of calcium that follows damage to the plasma membrane annexin A6 is recruited to multivesicular bodies (MVBs) and likely serves to tether the MVBs to the plasma membrane causing a concomitant release of exosomes.1. In the first sentence of the Discussion, the authors specify that their results suggest that exosome secretion is coupled to calcium-dependent plasma membrane repair in HCT116 cells and indeed they present strong evidence that this is the case. Although the authors did use U2-OS cells to investigate the role of annexin A6 domains, it is not clear if this process is restricted to HCT116 cancer cells or a more general phenomenon for both normal and neoplastic cells. This reviewer is not suggesting that every experiment needs to be performed in multiple cell lines but rather that a few key experiments be repeated in at least one other cell line, perhaps normal epithelial cells or fibroblasts. This would strengthen the manuscript.

We thank the reviewer for their suggestion. We generated a HEK293T reporter cell line that expresses FLAG-Nluc-CD63 under control of the low-expression L30 promoter. We then assessed exosome secretion upon addition of increasing amounts of SLO, in the absence or presence of calcium. We observed exosome secretion in this new HEK293T reporter cell line in a SLO dose- and calcium-dependent manner and have included these results as the new Figure 2 —figure supplement 1B.

2. The authors state (sentence 262, page 10) that "…ANXA6(FL) was localized to the plasma membrane and to intracellular vesicles (Figure 6B)" and also note (sentence 268, page 11) that "…ANXA6(C) was observed on intracellular vesicles". What are these intracellular vesicles? Are they referring to intracellular compartments such as MVBs or are they referring to the intraluminal vesicles inside MVBs that are destined to be released as exosomes? From the model (Figure 7), it is suggested that annexin A6 binds to MVBs so presumably, the implication from Figure 6 is that annexin A6 localizes to MVBs following ionomycin treatment. But that is not demonstrated. It would improve the quality of the data in Figure 6 if cells labeled for both annexin A6 and CD63 (or other MVB marker) were imaged to see the extent of co-association with MVB structures.

We have included additional imaging experiments in the new Figure 6 that demonstrate that a portion of ANXA6 is recruited to CD63-positive compartments upon ionomycin treatment. These data, taken together with our results from Figure 3C and 3D (in which we identified ANXA6 to be bound in a calcium-dependent manner to endogenous CD63-Nluc-positive MVBs immunopurified using an Nluc antibody), lead us to propose our current model in which ANXA6 is recruited to CD63positive membrane compartments following ionomycin treatment.

3. Please clarify if annexin A6 itself is present in/on exosomes themselves or is absent from these.4. Related to the question above, as annexin A6 can localize to the plasma membrane in ionomycin-treated cells, this would imply that microvesicles that bud off from the plasma membrane may also contain annexin A6. Thus, it is possible that annexin A6-positive microvesicles are released after plasma membrane damage or perturbation of the intracellular calcium level. The authors should comment on this possibility. The authors state that in response to ionomycin treatment, they observe the budding of plasma membrane-derived vesicles. If annexin A6 is present outside the cell, where does it partition after iodixanol density gradient fractionation – with CD63 or with annexin A2 and FLOT2?5. Also related to the questions above, the authors use their endogenous CD63-Nluc system to convincingly demonstrate that specifically exosome secretion is enhanced as a consequence of plasma membrane repair but what about other types of extracellular vesicles such as plasma membrane-derived microvesicles? Is the release of microvesicles also enhanced by the plasma membrane repair process?

We thank the reviewer for these great suggestions. We are currently conducting follow up research regarding the molecular mechanisms underlying the release of annexin-positive microvesicles upon plasma membrane disruption, and suggestions 3-5 will be included in that future work.

6. Although not directly addressed in the Discussion, this reviewer is left with the impression that the authors are hinting that exosome secretion is more a byproduct of plasma membrane repair rather than a means of intercellular communication. In other words, the cell needs the membrane material from the MVB to patch and repair holes in the plasma membrane and exosome ejection from the cell is a secondary (perhaps even irrelevant) consequence. Obviously, these two possibilities are not mutually exclusive. Nevertheless, the authors are encouraged to speculate about which possibility they favor and how their findings might change our understanding of the cell biology of exosome secretion.

We have added additional commentary to paragraph 4 of the Discussion section to speculate about our findings in relation to the field.

7. The authors might consider placing their findings in the context of Annexin A11 acting as a tether for the transport of RNA granules (Y-C Liao et al., Cell 179:147-164, 2019).

We have included an additional sentence within the discussion that places our findings in the context of this previous work.